# Deubiquitinating enzyme mutagenesis screens identify a USP43-dependent HIF-1 transcriptional response

Tekle Pauzaite [iD], Niek Wit [iD], Rachel V Seear & James A Nathan [iD] [✉]

## Abstract

The ubiquitination and proteasome-mediated degradation of Hypoxia Inducible Factors (HIFs) is central to metazoan oxygen-sensing, but the involvement of deubiquitinating enzymes (DUBs) in HIF signalling is less clear. Here, using a bespoke DUBs sgRNA library we conduct CRISPR/Cas9 mutagenesis screens to determine how DUBs are involved in HIF signalling. Alongside defining DUBs involved in HIF activation or suppression, we identify USP43 as a DUB required for efficient activation of a HIF response. USP43 is hypoxia regulated and selectively associates with the HIF-1α isoform, and while USP43 does not alter HIF-1α stability, it facilitates HIF-1 nuclear accumulation and binding to its target genes. Mechanistically, USP43 associates with 14-3-3 proteins in a hypoxia and phosphorylation dependent manner to increase the nuclear pool of HIF-1. Together, our results highlight the multifunctionality of DUBs, illustrating that they can provide important signalling functions alongside their catalytic roles.

**Keywords** Hypoxia; USP43; HIF; Deubiquitination; DUB
**Subject Categories** Post-translational Modifications & Proteolysis; Signal Transduction

## Introduction

The ability of organisms to sense and adapt to varying oxygen gradients is conserved across species. In metazoans, oxygen-sensing is principally governed by HIFs, which facilitate an adaptive transcriptional programme to respond to reduced oxygen availability (Ivan and Kaelin, 2017; Kaelin and Ratcliffe, 2008; Schofield and Ratcliffe, 2004; Semenza, 2012). The oxygen-sensitive nature of the HIF pathway relates to prolyl hydroxylation of the HIF-α subunit (principally HIF-1α or HIF-2α). When oxygen is abundant, the HIF-α subunit undergoes prolyl hydroxylation at two conserved residues by the prolyl hydroxylases (PHDs or EGLNs), which prime HIF-α for ubiquitination by the Von Hippel–Lindau (VHL) E3

ligase, leading to its rapid proteasome-mediated degradation (Bruick and McKnight, 2001; Epstein et al, 2001; Jaakkola et al, 2001; Maxwell et al, 1999). When the oxygen supply is limited, PHDs are inactivated, and HIF-α is stabilised, thereby allowing the formation of the stable HIF-α/HIF1β heterodimeric transcriptional complex, which binds to hypoxia-responsive elements (HREs) at HIF-responsive genes.

The dominance of VHL-mediated ubiquitination in controlling HIF-α stability is evident from extensive genetic studies of *VHL* mutations, and compounds inhibiting VHL enzymatic activity (Buckley et al, 2012; Cancer Genome Atlas Research, 2013; Frost et al, 2016; Latif et al, 1993; Maher and Kaelin, 1997; Maxwell et al, 1999; Mitchell et al, 2018; Turajlic et al, 2018). In all cases, VHL loss or inhibition leads to HIF-α stabilisation, even when oxygen levels are abundant. VHL-independent ubiquitination has also been observed (Ferreira et al, 2015; Flügel et al, 2012), and this may help regulate HIF-α levels when oxygen supply is limited, or to fine-tune the HIF response. However, while it is evident that ubiquitination of HIF-α is required for oxygen-sensing, the role of deubiquitination and deubiquitinating enzymes (DUBs) is less clear.

Several DUBs have been linked to reversing HIF-α stabilisation, including USP7, USP8, USP20, USP25, USP29, USP33, BAP1 and UCHL1 (Bononi et al, 2023; Gao et al, 2021; Goto et al, 2015; Li et al, 2005; Nelson et al, 2022; Troilo et al, 2014; Wu et al, 2016; Zhang et al, 2022). However, HIF-α deubiquitination does not generally overcome the dominant action of VHL when oxygen is present, and has mostly been observed by overexpression of the DUB, re-oxygenation experiments, or in VHL-deficient kidney cancer cells (Hong et al, 2020). It is also important to consider that DUBs may have roles outside of antagonising VHL, as demonstrated for Cezanne (OTUD7B) and PAN2 (USP52) (Bett et al, 2013; Bremm et al, 2014; Moniz et al, 2015).

Here, we take an unbiased approach to explore the contribution of DUBs to HIF signalling. Using CRISPR/Cas9 mutagenesis screens and a dynamic fluorescent reporter that provides a robust readout of HIF signalling, we conduct both suppressor and activator screens to define the principal DUBs involved in HIF regulation. Alongside exploring the involvement of DUBs in HIF regulation, our screening approach identified USP43, a poorly characterised DUB that was required for activation of the HIF-transcriptional response. RNA-seq corroborated the involvement of

Cambridge Institute of Therapeutic Immunology & Infectious Disease (CITIID), Jeffrey Cheah Biomedical Centre, Department of Medicine, University of Cambridge, Cambridge CB2 0AW, United Kingdom. ✉E-mail: jan33@cam.ac.uk

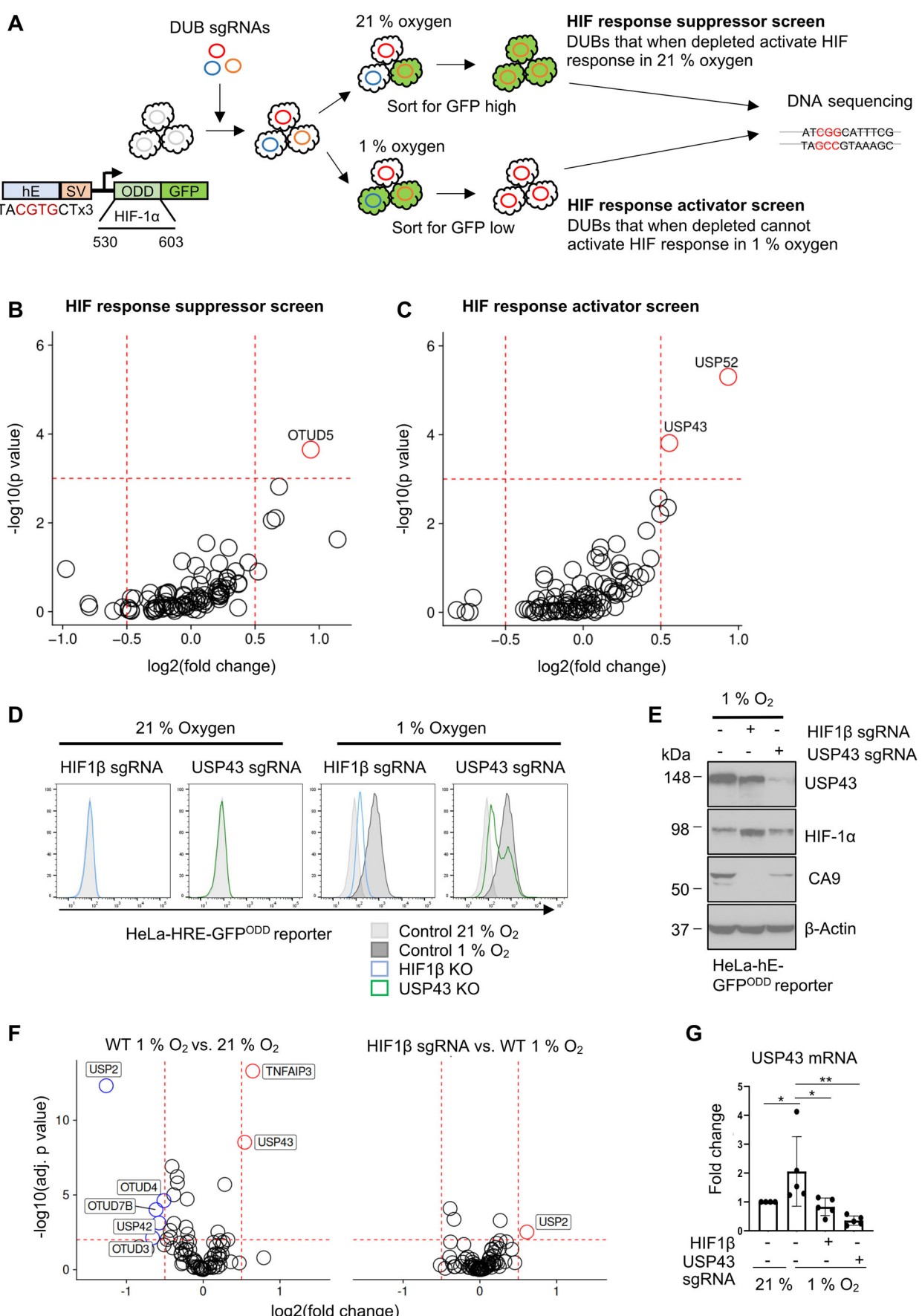

**Figure 1.  Mutagenesis screens identify USP43 as a regulator of the HIF response.**

(A) Schematic of the pooled DUB sgRNA library screen. Diagram of HIF-HRE-GFP$^{ODD}$ reporter in HeLa-Cas9 cells (left). For the suppressor screen, HeLa HRE-$^{ODD}$GFP reporter cells were transduced with a DUB sgRNA library, and sorted for a GFP high population using flow cytometry. For the activator screen, cells were incubated in 1% oxygen for 24 h and sorted for GFP low-expressing cells. (B, C) Comparative bubble plots of HIF suppressor screen (B), or activator screen (C). DNA was extracted, sgRNAs identified by Illumina MiniSeq, and compared between sorted and unsorted libraries. Unadjusted $P$ value calculated using MaGECK robust rank aggregation (RRA). (D) HeLa HRE-GFP$^{ODD}$ reporter expression in mixed knockout populations (denoted as sgRNA) of HIF1β (light blue) or USP43 (green). Reporter levels were analysed by flow cytometry for GFP 488 nm following incubation in 21 or 1% oxygen (16 h). (E) Immunoblot of HIF1β or USP43 depleted HeLa HRE-GFP$^{ODD}$ reporter cells after 16 h of 1% oxygen. Representative of three biological replicates. (F) RNA-seq analysis of HeLa control or a mixed KO population of HIF1β KO cells, treated with either 21 or 1% oxygen for 16 h. $n = 3$ biologically independent samples. Volcano plots of differential mRNA expression in control 1 versus 21% oxygen (left), or HIF1β KO cells compared with control cells in 1% oxygen (right). A log2(fold change) of >0.5 or <−0.5, and −log10 ($p$ value) of >2.5 was selected as the significance level. (G) Quantitative RT-PCR (qPCR) of USP43 mRNA in HeLa cells. A mixed KO population of HIF1β or USP43 HeLa cells were incubated in 1 or 21% oxygen for 16 h. $n = 5$ biologically independent samples. Mean ± sd. 21% $O_2$ vs. 1% $O_2$ *$P = 0.0427$, control vs. HIF1β sgRNA 1% $O_2$ *$P = 0.0132$, control vs. USP43 sgRNA 1% $O_2$ **$P = 0.0012$, one-way ANOVA. Source data are available online for this figure.

USP43 in HIF signalling, with USP43 as one of only two DUBs that is upregulated in hypoxia in an HIF-dependent manner. Functionally, USP43 depletion decreases the activation of HIF-1 target genes, without altering HIF-2 signalling. This selectivity of USP43 for HIF-1 was not due to altered HIF-α protein or mRNA levels. Instead, we show that USP43 associates with HIF-1α and promotes the nuclear accumulation of HIF-1 and the chromatin binding of the transcriptional complex to specific HIF-1 target genes. Remarkably, USP43 facilitates HIF-1 nuclear accumulation through hypoxia-sensitive recruitment of 14-3-3 proteins, rather than its deubiquitinase activity. Therefore, this work demonstrates the ability of a DUB to influence a transcriptional programme and highlights the broader role of DUBs in intracellular signalling pathways.

## Results

### Mutagenesis screens identify USP43 as a regulator of the HIF response

To unbiasedly identify deubiquitinating enzymes (DUBs) involved in HIF regulation we used a phenotypic HIF functional screen using a pooled DUB CRISPR sgRNA library. The screens were performed in HeLa cells expressing a well-validated HIF reporter (HIF$^{ODD}$GFP reporter), which we have previously used to interrogate the regulation of the HIF response (Bailey et al, 2020; Burr et al, 2016; Ortmann et al, 2021). This reporter has the advantage that it provides a dynamic readout to both endogenous HIF stability and transcriptional activation via binding to a triplicate HRE (Fig. 1A). The sub-pooled DUB sgRNA library was generated as part of a "ubiquitome library", encoding 10 sgRNAs per gene (Menzies et al, 2018). We performed two opposing mutagenesis screens, aimed at identifying DUBs that either activate or suppress the HIF response (Fig. 1A; Dataset EV1). OTUD5 was significantly enriched in the suppressor screen (Fig. 1B), whereas both USP43 and USP52 (also known as PAN2) were enriched in the activator screen (Fig. 1C). USP52 has been implicated in HIF regulation, through its global effect on RNA stability, and we validated this finding (Fig. EV1A) (Bett et al, 2013; Wolf and Passmore, 2014). OTUD5 (also known as DUBA) has been implicated in multiple cell pathways, including mTOR regulation (Cho et al, 2021), and OTUD5 depletion only had a marginal change in HIF$^{ODD}$GFP reporter levels in 21% oxygen (Fig. EV1B). However, the involvement of USP43, a poorly

characterised DUB, in the HIF response was unknown, and we confirmed that USP43 depletion prevented full activation of the HIF$^{ODD}$GFP reporter in 1% oxygen (Figs. 1D,E and EV1A).

We validated the findings of the HIF activator screen in two non-cancer cell lines, HKC-8 (renal tubular epithelial cells) and RPE-1 (retinal pigment epithelial cells), where HIF responses have been well-validated (Cristante et al, 2023; Smythies et al, 2019). Endogenous HIF-1α levels or activation of the HIF-1 target gene, Carbonic Anhydrase 9 (CA9), were measured by flow cytometry following sgRNA depletion of the top 10 DUBs in the screen (Fig. EV1C–H). USP43 depletion reduced CA9 levels in both cell types, without altering HIF-1α steady-state levels, consistent with our findings using the HIF$^{ODD}$GFP reporter (Fig. EV1D–H). Four of the other top hits validated in either or both cell lines—USP52, JOSD1, UCHL5 and OTUB1.

We reasoned that DUBs that are regulatory in HIF signalling might be reciprocally regulated by HIFs, as observed by other components of the HIF pathway (Minamishima et al, 2009; Ortmann et al, 2021). Therefore, using RNA-seq (Figs. 1F,G and EV1I; Dataset EV2), we analysed which DUBs are transcriptionally regulated in hypoxia, and whether this is dependent on HIFs. HeLa cells exposed to 1 or 21% oxygen for 16 h demonstrated that few DUBs had altered expression in hypoxia (Fig. 1F). OTUD7B (also known as Cezanne) was the top-downregulated DUB in hypoxia, consistent with its known involvement in non-canonical HIF stability (Bremm et al, 2014; Moniz et al, 2015). USP43 was transcriptionally upregulated in hypoxia in HeLa cells (Fig. 1F,G), and we also observed hypoxic induction of USP43 in A549 and HKC-8 cells, but not in MCF7 cells (Fig. EV1J). The hypoxic induction of USP43 was not observed in HIF1β null cells indicating that USP43 expression was HIF inducible (Fig. 1G). As USP43 was both a top hit on the screens and was itself regulated in a HIF-dependent manner, we focused our further studies on the actions of this DUB.

### USP43 is required for activation of a HIF-1 response

The validation of the activator screen in HKC-8 and RPE-1 cells implicated USP43 in endogenous HIF signalling. To examine this further, we generated mixed knockout (KO) populations of USP43 using sgRNA in HeLa, A549, and MCF7 cells. All showed reduced protein levels of the HIF-1 target gene CA9 (Fig. 2A). The reduction in CA9 was not to the same extent as completely ablating the HIF pathway by HIF1β loss, but CA9 levels were reduced without any appreciable difference in protein levels of the two main HIF-α

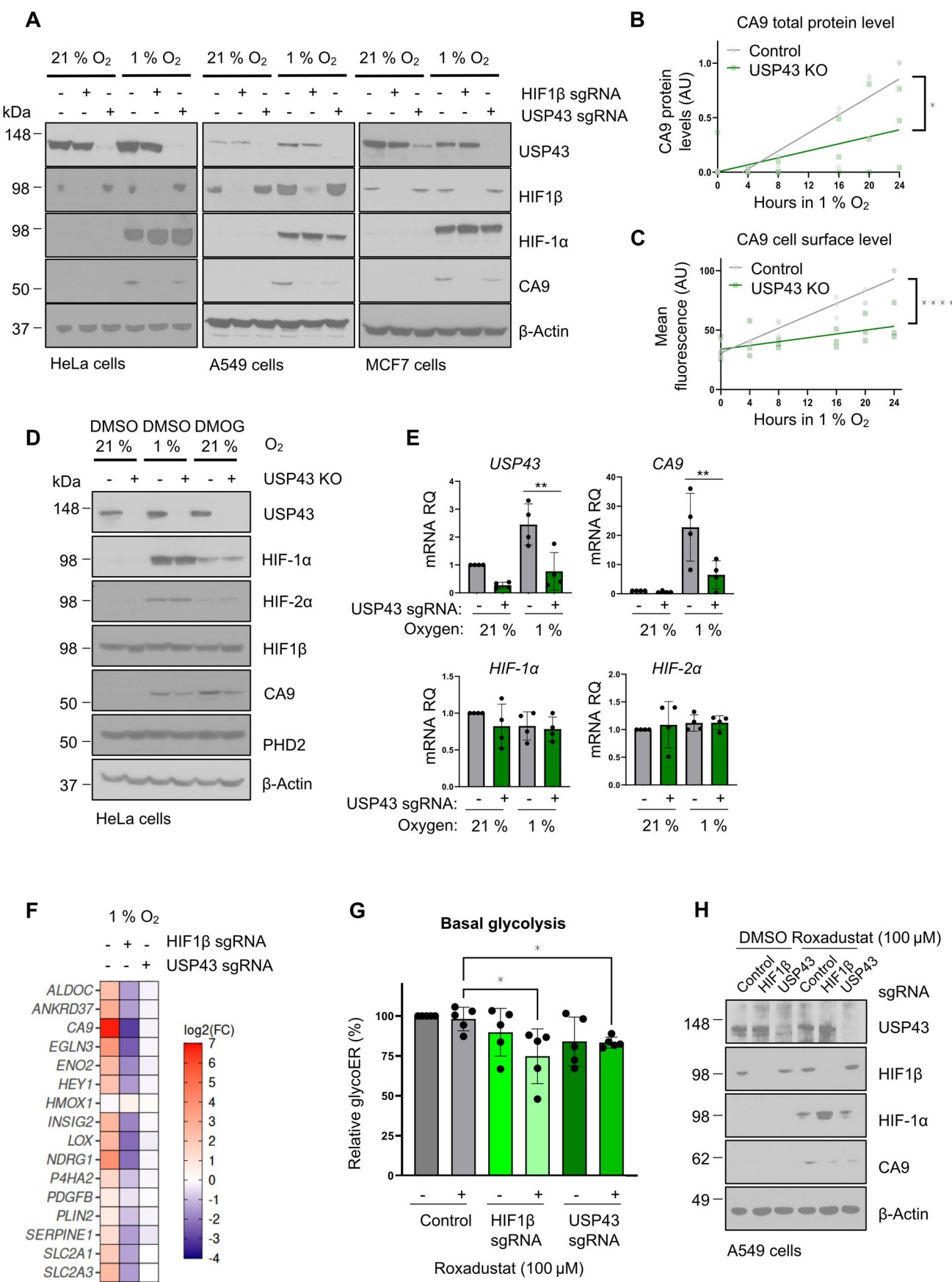

**Figure 2.   USP43 is required for activation of a HIF-1 response.**

(A) USP43 depletion in different cell types. Mixed KO populations of USP43 or HIF1β HeLa, A549 and MCF7 cells were generated. Cells were incubated in 21 or 1% oxygen for 16 h and immunoblotted. Representative of three biological replicates. (B, C) Quantification of total (B) and cell surface, (C) CA9 protein levels in control or USP43 null (KO denotes null clone) HeLa cells after 0–24 h of 1% oxygen. (B) Total protein levels measured by immunoblot densitometry relative to a β-actin control. $n = 3$ biological replicates, $*P = 0.0322$, two-way ANOVA. (C) Cell surface CA9 levels are measured by flow cytometry, and the mean fluorescence intensity is shown. $n = 3$ biological replicates, $****P < 0.0001$ control vs. USP43 KO, two-way ANOVA. Representative immunoblots and flow cytometry are shown in Fig. S1C–F. (D) Immunoblot of HeLa-Cas9 or USP43 null clone after 16 h of 1% oxygen or 24 h of 1 mM DMOG treatment. Representative of three biological replicates. (E) qPCR of *USP43*, *CA9*, *HIF-1α*, or *HIF-2α* in control or USP43 depleted HeLa cells following incubation in 1 or 21% oxygen for 16 h. $n = 4$ biologically independent samples, mean ± sd. *USP43*: control vs. USP43 sgRNA 1% O$_2$ $**P = 0.0011$. *CA9*: control vs. USP43 sgRNA 1% O$_2$ $**P = 0.0086$. One-way ANOVA. (F) Heat map representing mRNA expression of validated HIF target genes analysed by RNA sequencing in control, HIF1β or USP43 mixed KO HeLa cells. (G, H) Bioenergetic analysis. Control, HIF1β depleted or USP43 depleted A549 cells were incubated with DMSO vehicle control or Roxadustat (0.1 mM) for 24 h. Cells were analysed using a Seahorse XF analyser by performing a glycolytic rate assay (G). The protein efflux rate derived from glycolysis (glycoPER) as a percentage of the total were calculated using WAVE version 2.6.1. $n = 5$, biologically independent samples, mean ± sd, one-way ANOVA. Control vs. HIF1β sgRNA + Rox $*P = 0.0348$. Control vs. USP43 sgRNA +Rox $*P = 0.0269$. Immunoblot of samples used for the used for Seahorse analysis (H). Control, HIF1β or USP43 depleted A549 cells, incubated with DMSO or Roxadustat (0.1 mM) for 24 h. Source data are available online for this figure.

isoforms (HIF-1α, HIF-2α) or HIF1β (Fig. 2A; Appendix Fig. S1A,B). USP43 clonal loss both delayed the HIF-driven CA9 induction and decreased the amplitude of the HIF response (Fig. 2B,C; Appendix Fig. S1C–F). Similar findings were observed using dimethyloxalylglycine (DMOG), a broad-spectrum inhibitor of PHDs and 2-oxoglutarate-dependent dioxygenases (Fig. 2D; Appendix Fig. S1B), indicating that USP43 was likely involved in HIF signalling downstream of prolyl hydroxylation. Quantitative PCR analysis in A549, HeLa, and MCF-7 cells confirmed that USP43 depletion decreased the expression of selected HIF target genes but not expression of HIF-1α or HIF-2α (Figs. 2E and EV2A–C). USP43 depletion did not alter apoptosis (Fig. EV2D) or growth after prolonged culture in 1% oxygen (Fig. EV2E).

To further understand the role of USP43 in HIF-transcriptional activation, we undertook RNA-seq analysis in HeLa control or USP43-depleted cells following 16 h of incubation in 21 or 1% oxygen (Figs. 2F and EV1I). HIF1β depletion was used as a control for abrogating the HIF response. Principal component analysis (PCA) showed that all experimental conditions clustered similarly in 21% oxygen, indicating that USP43 or HIF1β loss did not globally alter transcription when oxygen was present (Fig. EV1I). USP43 depletion also clustered with the control HeLa cells exposed to 1% oxygen, rather than the HIF1β deficient cells where the HIF-transcriptional response is completely ablated. These findings suggested that USP43 loss either had a moderate effect on HIF target gene expression or altered a subset of HIF target genes. Indeed, the expression analysis of HIF target genes showed a partial reduction in transcript levels when compared with HIF1β KO in 1% oxygen, and interestingly, USP43 loss altered the transcription of genes involved in glycolysis (Fig. 2F). Bioenergetic analysis using a Seahorse confirmed that USP43 loss decreased basal glycolysis following PHD inhibition with Roxadustat (Fig. 2G,H). Therefore, USP43 loss perturbs the HIF response in a selective manner, and can impair the HIF-driven shift to glycolysis.

Given that USP43 only regulated a subset of HIF-target genes, we investigated whether USP43 had any specificity towards HIF-α isoforms. Using HeLa HIF$^{ODD}$GFP reporter cells that were either HIF-1α or HIF-2α deficient (Ortmann et al, 2021), we observed that USP43 loss only altered HIF-1 transcriptional activation of the reporter (Fig. 3A,B). These findings were substantiated using HIF-1α or HIF-2α null HeLa cells, whereby we observed a reduction in the activation of HIF-1 but not HIF-2 target genes (Appendix Fig.

S2A–D). To further validate these findings, we examined the effect of USP43 loss in clear cell renal carcinoma cell (ccRCC) lines that have constitutive activation of both HIF-1 and HIF-2 (RCC4 cells, *HIF-1α+*, *HIF-2α+* and *VHL−*), or only encode HIF-2α (786-0 cells, *HIF-1α−*, *HIF-2α+* and *VHL−*). USP43 depletion reduced protein, transcript, and cell surface levels of CA9 in RCC4 cells (Fig. 3C–E), but did not influence HIF-2α driven transcription in 786-0 cells (Fig. 3F,G). Therefore, USP43 shows specificity towards the HIF-1α isoform.

## USP43 is an active DUB that associates with the HIF-1 complex

To understand how USP43 may facilitate a HIF-1 transcriptional response, we examined if USP43 associated with HIF-1α in hypoxia by incubating HeLa cells in 21 or 1% oxygen for 6 h and immunoprecipitating endogenous USP43 or HIF-1α. USP43 bound to HIF-1α in hypoxia, and depletion of HIF1β did not prevent the association between endogenous USP43 and HIF-1α (Figs. 4A,B and EV3A,B). No association of endogenous or overexpressed USP43 with HIF-2α was observed (Fig. EV3A–C), consistent with the HIF-1α specificity observed in the ccRCC lines. The association of USP43 with HIF-1α was also observed in other cell lines (A549 and MCF7 cells) (Fig. EV3D,E). Importantly, the interaction between HIF-1α and USP43 did not prevent the formation of a HIF heterodimer, as HIF1-α and HIF1β still associated in USP43 null HeLa cells (Fig. EV3F), and neither USP43 depletion nor overexpression altered HIF-1α degradation kinetics using cyclo-heximide chase assay of HIF-1α degradation (Fig. EV3G–J).

Given the lack of effect of USP43 on HIF-1α stability and that the deubiquitinase activity of USP43 has not been well studied, we considered whether USP43 was an active DUB or a pseudoDUB as observed for other USP proteins (Bett et al, 2013; Wolf and Passmore, 2014). We generated mutations of the putative catalytic triad of USP43 (nucleophilic cysteine 110 and proton acceptor histidine 668) based on the Alphafold predictions (Jumper et al, 2021) (Fig. 4C,D), and used a biotinylated DUB probe (Biotin-ANP-Ub-PA) that covalently binds the active cysteine to investigate DUB activity. Wildtype USP43 was able to bind the ubiquitin probe, demonstrating that USP43 is catalytically active (Fig. 4E), but mutation of cysteine 110 to alanine (C110A), with or without a histidine 668 to arginine (H668R) mutation rendered USP43 catalytically inactive.

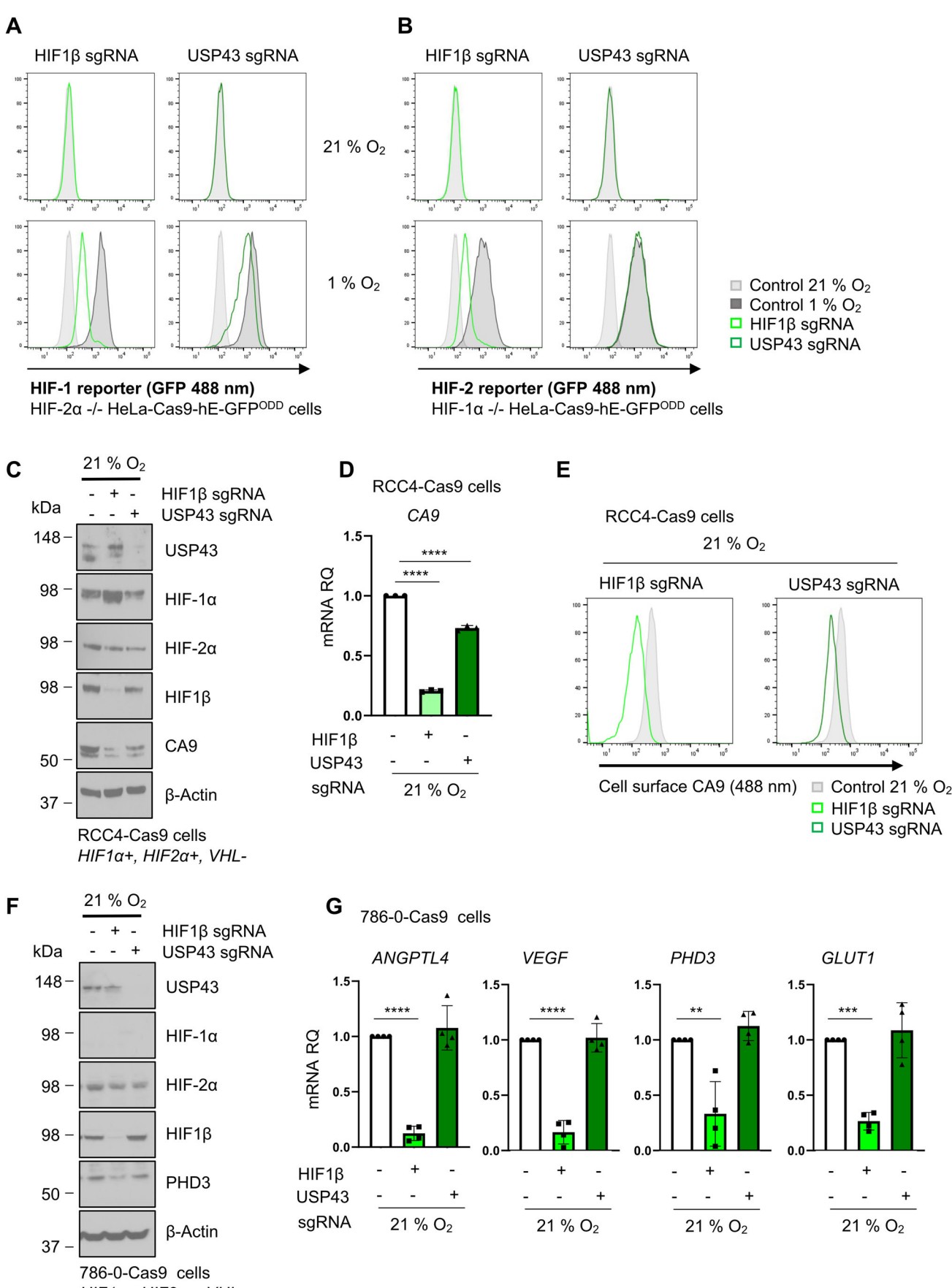

**Figure 3.  USP43 is specific for the HIF-1 complex.**

(A, B) Mixed KO populations of either USP43 or HIF1β in HIF-1 reporter cells (HIF-2α null HeLa HRE-$^{ODD}$GFP) (A) or HIF-2 reporter cells (HIF-1α null HeLa HRE-$^{ODD}$GFP) (B) were generated, treated with 1 or 21% oxygen for 24 h, and reporter activity measured by flow cytometry. Representative of three biological replicates. (C–E) USP43 depletion in RCC4 cells. Mixed KO populations of USP43 or HIF1β were generated in RCC4 cells stably expressing Cas9. HIF complex component levels and CA9 levels were measured by immunoblot (C). CA9 expression (D) and CA9 cell surface protein levels (E) were analysed by qPCR and flow cytometry respectively. Representative of three biological replicates. Mean ± sd. Control vs. HIF1β sgRNA ****$P < 0.0001$; control vs. USP43 sgRNA ****$P < 0.0001$, one-way ANOVA (D). (F, G) USP43 depletion in 786-0 cells. USP43 mixed KO populations were generated in 786-0 cells stably expressing Cas9. HIF complex components and HIF-2 target genes were analysed by immunoblot (F) or qPCR (G) as previously described. $n = 4$ biological replicates, mean ± sd. ANGPTL4: control vs. HIF1β ****$P < 0.0001$. VEGF: control vs. HIF1β ****$P < 0.0001$. PHD3: control vs. HIF1β **$P = 0.0012$. GLUT1: control vs. HIF1β ***$P = 0.001$. One-way ANOVA. Source data are available online for this figure.

To determine the involvement of USP43 deubiquitinase activity in HIF signalling, we tested if reconstitution of the catalytically active or inactive USP43 restored HIF$^{ODD}$GFP reporter or CA9 levels following USP43 depletion. USP43 expression restored HIF activity in HeLa and A549 USP43 deficient cells (CRISPR/Cas9 KO or siRNA) in a dose-dependent manner, similar to control conditions (Fig. 4F–I; Appendix Fig. S3A–D). Surprisingly, the USP43$^{C110A+H667R}$ mutant also partially restored HIF reporter activity and CA9 levels (Fig. 4F–I), suggesting that while USP43 enzymatic activity may be involved, deubiquitination itself was not essential for the effect of USP43 on HIF-mediated transcription.

## USP43 regulates HIF-1α nuclear accumulation

We postulated that USP43 must act downstream of HIF-1α stabilisation, and was involved in either subcellular localisation of HIF-1, or the interaction of the transcription factor with chromatin. We therefore examined where USP43 was localised within the cell, and if this altered under hypoxic conditions. Immunofluorescence of endogenous USP43 was not possible, but using HA-USP43 we observed that USP43 was predominantly localised within the cytosol in both 21 and 1% oxygen (Fig. 5A). However, some USP43 localised within the nucleus, and this was markedly increased in hypoxia (Fig. 5A,B). We confirmed that endogenous USP43 was also localised to the nucleus using biochemical fractionation of HeLa cell lysates (Figs. 5C and EV4A,B). This translocation of USP43 resulted in a rapid increase in USP43 within the chromatin fraction and corresponded with the chromatin-associated HIF-1α, and still occurred when the HIF-1 complex was absent (Fig. 5C and EV4C,D). Using a more stringent isolation of chromatin, involving formaldehyde crosslinking and urea extraction (Kustatscher et al, 2014), we still observed USP43 in the chromatin fraction, and noticed that HIF-1α enrichment in the chromatin fraction was reduced in USP43 KO cells and increased when USP43 was overexpressed (Fig. 5D,F). Therefore, we examined if USP43 altered the kinetics of HIF-1α localisation to the nucleus in hypoxia. USP43 deficient HeLa cells, incubated in 1% oxygen for 20, 40 and 60 min showed reduced HIF-1α levels on chromatin, while USP43 over-expression increased HIF-1α levels in the chromatin fraction (Figs. 5E,G and EV4C,D). Overexpression of the USP43 catalytic inactive mutant (USP43$^{C110A+H667R}$) did not increase HIF-1α in the chromatin fraction (Fig. EV4E,F). Therefore, while both catalytic and non-catalytic activity of USP43 may be involved, our data are consistent with a role for USP43 in HIF nuclear localisation or retention.

Given that USP43 levels perturbed the kinetics of the abundance of HIF-1α on chromatin, we next determined if this altered the binding of HIF-1α to known target gene loci in HeLa cells. Chromatin immunoprecipitation (ChIP)-PCR of HIF-1α binding showed that USP43 depletion decreased HIF-1α binding to HREs

on the CA9 and PHD3 loci in hypoxia (Fig. 6A). The decrease in HIF-1α binding was not as substantial as HIF1β depletion, consistent with the functional consequences of USP43 loss described previously. USP43 loss also did not alter HIF-1α binding to the VEGF HRE, similar to the lack of effect of USP43 depletion on VEGF using RNA-seq. Comparable findings were observed using HIF1β ChIP-PCR (Fig. 6B), indicating that USP43 depletion altered the binding of the HIF-1 complex at selected HIF-1 loci. Interestingly, these findings were distinct from other known co-activators involved in the regulation of HIF target genes, such as the SET1B histone methyltransferase, as we observed no changes in binding of the HIF-1 complex to chromatin with SET1B loss, but both USP43 and SET1B loss still resulted in reduced histone 3 lysine 4 tri-methylation (H3K4me3) at the CA9 locus (Fig. 6C,D). Moreover, we did not observe any changes in Histone 2B lysine120 ubiquitination (H2BK120Ub) following USP43 loss, nor an association between USP43 and H2BK120Ub, in contrast to a prior report implicating USP43 in the regulation of this activating epigenetic mark (He et al, 2018) (Appendix Fig. S4A,B).

## USP43 associates with 14-3-3 proteins to regulate HIF-1 signalling

The ability of USP43 levels to influence HIF-1α levels on chromatin suggested that a further protein or signal was involved. We, therefore, used the global DUB mass spectrometry interactome (Sowa et al, 2009) to find potential USP43 binding partners. USP43 was the only DUB shown to strongly interact with 14-3-3 proteins (Sowa et al, 2009), and given the importance of 14-3-3 in nuclear-cytoplasmic transport (Brunet et al, 2002) and potential HIF-1 interactions (He et al, 2018; Xue et al, 2016), we hypothesised that USP43 may regulate HIF-1 localisation through the recruitment of 14-3-3 proteins. We confirmed that USP43 binds 14-3-3 proteins, using overexpressed HA-USP43 and endogenous USP43 in HeLa cells (Fig. 7A,B), consistent with the DUB mass spectrometry interactome (Sowa et al, 2009). Moreover, we found that 14-3-3 proteins bound USP43 more strongly in 1% oxygen (Fig. 7B,C).

14-3-3 proteins are well known to interact with phosphorylated proteins. Consistently, we showed that USP43 is phosphorylated using Lambda protein phosphatase (Lambda PP) and a gel shift assay (Fig. EV5A). We also observed that immunoprecipitation with a phospho-Serine (pSer) 14-3-3 motif antibody pulled down USP43, and importantly, this interaction was hypoxia-dependent (Fig. EV5B). Moreover, the interaction between 14-3-3 and USP43 could be replicated using Culyculin A as a phosphatase inhibitor (Resjö et al, 1999), and conversely, prevented by Lambda PP (Fig. EV5C).

We examined if the USP43 and 14-3-3 interaction was important for the recruitment of HIF-1α, and observed that

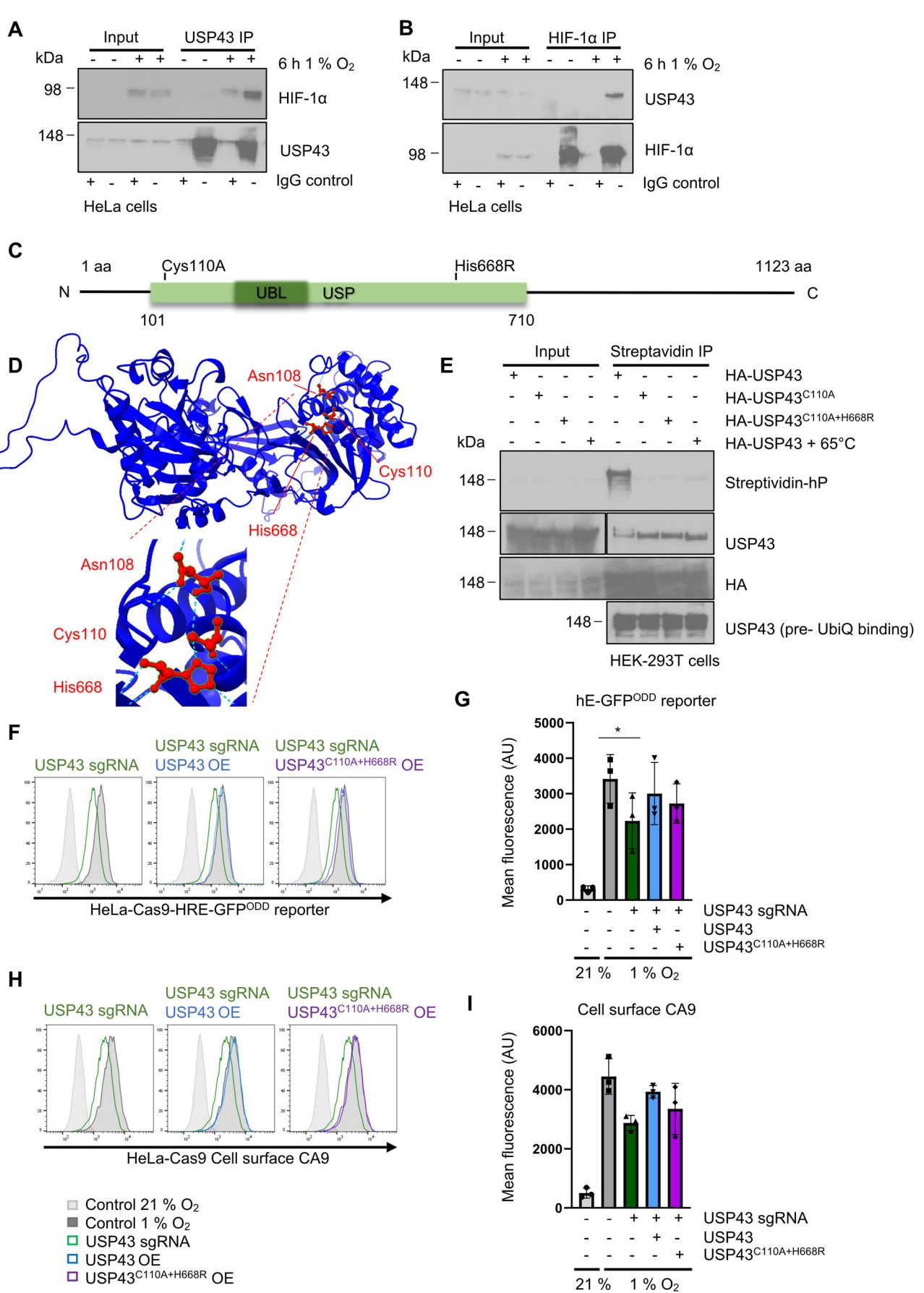

**Figure 4. USP43 is an active DUB that associates with the HIF-1 complex.**

(A) Endogenous USP43 was immunoprecipitated in HeLa cells grown at 21 or 1% oxygen for 6 h. Samples were immunoblotted for HIF-1α. Rabbit IgG was used as a control for non-specific binding. Representative of three biological replicates. (B) As in (A), endogenous HIF-1α was immunoprecipitated and samples were immunoblotted for endogenous USP43. (C) Schematic of USP43 protein sequence, with ubiquitin-specific protease (USP) and ubiquitin-like domains (UBL) shown. The positions of the active site nucleophile cysteine 110 (Cys110) and proton acceptor histidine 668 (His668) are indicated. (D) Predicted USP43 catalytic triad using AlphaFold (10.1093/nar/gkab1061) and ChimeraX-1.6.1.exe. The catalytic sites Cys110, His668 and Asn108 are labelled. (E) Immunoblot of DUB catalytic activity-based probe Biotin-ANP-Ub-PA (UbiQ-077) (1 μM) binding to the overexpressed HA-USP43, HA-USP43$^{C110A}$ or HA-USP43$^{C110A+H668R}$ in HEK293T cells. (F, G) HeLa HRE-$^{ODD}$GFP reporter cells or USP43 mixed population KO cells (dark green) reconstituted with HA-USP43 (blue) or HA-USP43$^{C110A+H668R}$ (purple) after 16 h of 1% oxygen treatment. HIF reporter activity was measured by flow cytometry (F) and quantified (G) using FlowJo10. Each dot represents the mean fluorescence intensity (AU at 488 nm) of a biological replicate. $n = 3$, biologically independent samples, mean ± sd. Control vs. USP43 sgRNA 1% O$_2$ *$P = 0.0107$, one-way ANOVA. (H, I) As for (F, G) but for CA9 cell surface levels in HeLa cells. $n = 3$, biologically independent samples, mean ± sd. Source data are available online for this figure.

USP43 loss decreased levels of HIF-1α associated with endogenous 14-3-3 (Fig. 7D,E). This interaction was not dependent on the formation of a HIF heterodimer, as 14-3-3 proteins bound to USP43 and HIF-1α in HIF1β null HeLa cells (Fig. EV5D). However, the association of USP43 with 14-3-3 was important for regulating HIF-1 signalling, as pan-siRNA-mediated depletion of 14-3-3 (targeting 6 of 7 isoforms) (Fig. EV5E) reduced activation of the fluorescent HIF reporter and endogenous CA9, without affecting HIF-1α protein and mRNA levels (Figs. 7F,G and EV5F,G).

To substantiate the involvement of 14-3-3 proteins, we attempted to map the 14-3-3 interaction site but with 168 potential binding sites identified (Madeira et al, 2015), further investigation of the specific 14-3-3 binding sites by site-directed mutagenesis was unfeasible. We, therefore, focused on which 14-3-3 isoform was involved. We generated sgRNAs targeting all seven 14-3-3 isoforms and observed the largest effect on HIF-1 signalling following depletion of zeta (14-3-3ζ) and epsilon (14-3-3ε) (Figs. 7G and EV5H). These findings were consistent with the anti-14-3-3 pan siRNA predominantly targeting 14-3-3ζ and 14-3-3ε (Figs. 7G EV5E,F). Furthermore, when we overexpressed HA-14-3-3ζ or 14-3-3ε, we found HA-14-3-3ζ bound more USP43 in 1% oxygen compared to 21% oxygen (Fig. EV5I,J).

If the binding of USP43 to 14-3-3 isoforms was important for HIF-1 signalling, we would expect a defect in HIF-1α binding to chromatin and that 14-3-3 and USP43 function within the same pathway. In support of this, we observed that pan siRNA-mediated depletion of 14-3-3 proteins reduced the levels of HIF-1α by chromatin fractionation (Figs. 7H and EV5K) and decreased the binding of HIF-1α to the HREs of *CA9* and *PHD3*, similarly to USP43 depletion (Fig. 7I). Lastly, we confirmed that both USP43 and 14-3-3 function in the same pathway by using siRNA-mediated depletion of 14-3-3 in HeLa USP43 null clones. 14-3-3 depletion did not further reduce HIF-1α binding to the *CA9* locus or H3K4me3 levels in the USP43 KO cells (Fig. 7J). Moreover, *CA9* mRNA expression and protein levels were reduced to the same extent in the USP43 null clones irrespective of the combined depletion of 14-3-3 (Figs. 7K,L and EV5L). Together, these data confirm that USP43 helps mediate the interaction between HIF-1α and 14-3-3ζ/ε, which in turn supports the chromatin localisation of HIF-1α and activation of HIF-1 target genes (Fig. 8).

## Discussion

Our approach of using a pooled DUB sgRNA library allowed us to unbiasedly screen for DUBs involved in the HIF response. This strategy proved powerful in identifying USP43 and its involvement in HIF-1 activation through associating with 14-3-3ζ/ε. It may seem surprising that USP43 catalytic activity was not absolutely required, but our findings highlight that DUBs may also serve important non-catalytic roles.

USP43 was found to strongly associate with 14-3-3 proteins in an unbiased mass spectrometry interactome (Sowa et al, 2009), and given the multifunctional roles of 14-3-3 proteins, this interaction likely helps facilitate USP43 function, or localise USP43 within the cell. The predominant effect of USP43 on the HIF-1 transcriptional response was through association with 14-3-3ζ/ε. USP43 has been reported to associate with other 14-3-3 isoforms, including a 14-3-3β/ε heterodimer (He et al, 2018). It is, therefore, plausible that USP43 interacts with different 14-3-3 proteins depending on the context. In addition, as 14-3-3 proteins interact with phosphorylated ligands (Fu et al, 2000), phosphorylation is likely to provide the recruitment signal, in keeping with our observation that Ser phosphorylated USP43 bound 14-3-3 proteins.

The role of 14-3-3 proteins in HIF signalling has not been well explored. 14-3-3ζ has been implicated in influencing HIF-1α expression (Tang et al, 2015; Xue et al, 2016), but we did not observe any changes in HIF-1α levels. Instead, the 14-3-3ζ/ε isoforms control the nuclear pool of the HIF-1 complex. The rapid changes in HIF-1α levels on chromatin suggest that the USP43/14-3-3 interaction with HIF may help with the initial nuclear localisation of the transcription factor. USP43 encodes a potential nuclear localisation motif that may be involved. Alternatively, 14-3-3 proteins may assist with the nuclear translocation due to their ability to enable free shuttling between the nucleus and cytoplasm (Brunet et al, 2002), or through ligand binding. Elucidating the dynamics and mechanisms of these 14-3-3/USP43 interactions and how they control HIF-1α nuclear import/export will be an important area of future work. Additionally, while our findings support phosphorylation as the signal for the association of USP43 with 14-3-3 proteins and HIF-1α, what drives this phosphorylation in hypoxia warrants further exploration.

We did not observe a clear role for USP43 catalytic activity, but it is possible that USP43 may serve dual functions, in both helping maintain the nuclear pool of HIF-1α via 14-3-3 recruitment, alongside a potential catalytic role of USP43 on chromatin. Recruiting a DUB via a transcription factor may facilitate local chromatin modifications via 14-3-3 (Winter et al, 2008) or histone deubiquitination (Peña-Llopis et al, 2012). While we did not observe any global change in H2BK120 ubiquitination following USP43 depletion, as previously reported (He et al, 2018), USP43 may remove other histone ubiquitin marks.

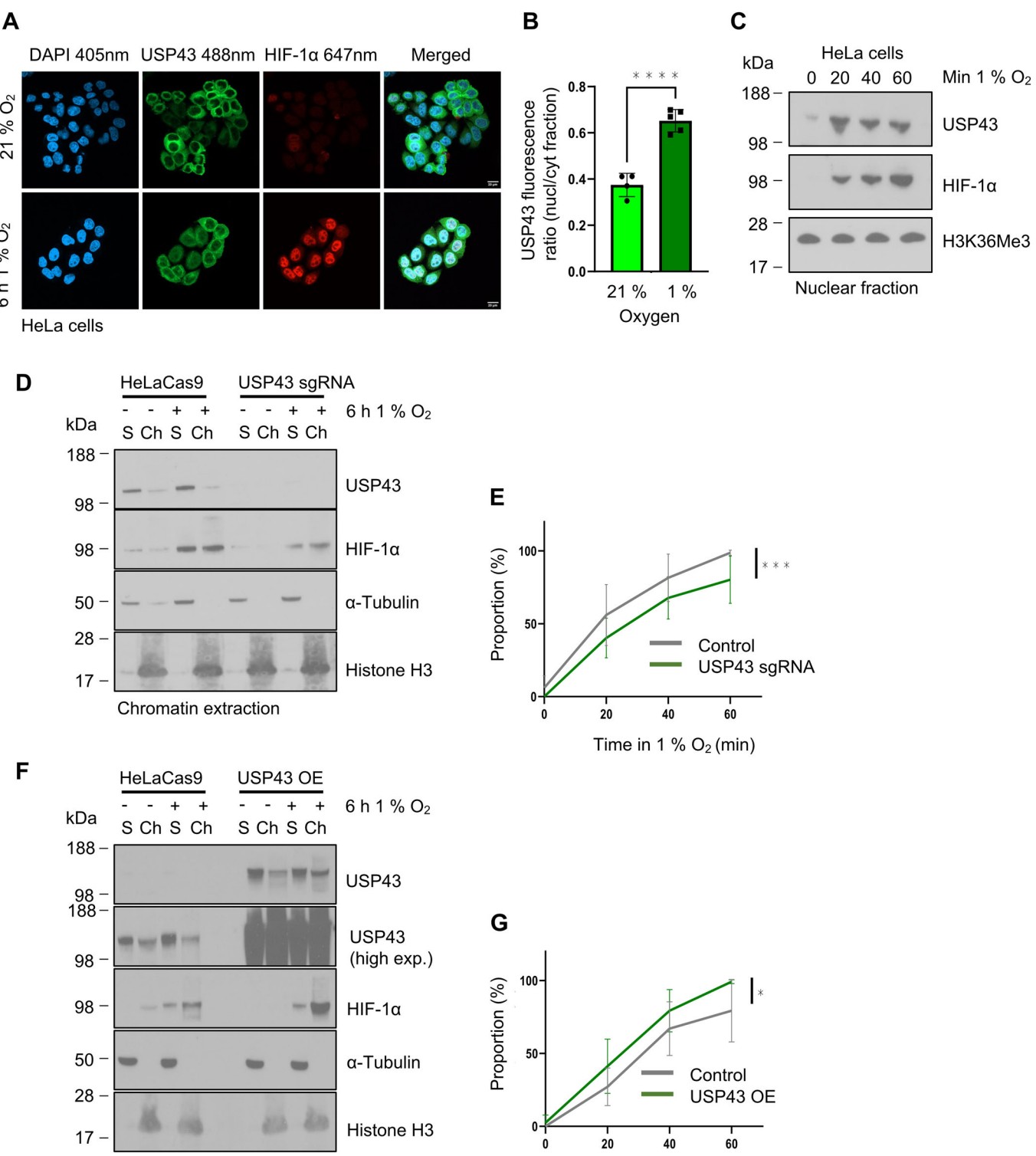

Deubiquitination as a mechanism to reverse the action of VHL has been observed (Gao et al, 2021; Li et al, 2005; Mennerich et al, 2019; Troilo et al, 2014; Wu et al, 2016), but the relative importance of individual DUBs remains unresolved. A single DUB that significantly counteracted VHL-mediated ubiquitination was not identified in our suppressor screen, as OTUD5 loss only had a very

marginal effect on the HIF$^{ODD}$GFP reporter in 21% oxygen. It is possible that there is redundancy or cell type specificity within DUBs reversing VHL-mediated ubiquitination. Tissue specific responses have been observed (e.g. glioma stem cells and pancreatic cancer cells) (Nelson et al, 2022; Zhang et al, 2022), potentially explaining why we did not detect DUBs that have been previously

**Figure 5. USP43 regulates HIF-1α nuclear accumulation.**

(A, B) Confocal immunofluorescence microscopy of USP43 overexpressing HeLa cells stained for USP43 (488 nm) and HIF-1α (647 nm). Cells were grown at 21 or 1% oxygen for 6 h on coverslips prior to fixation. Representative images (A) and quantification (B) are shown. Each point represents a biological replication of ~25 cells acquired and analysed as one image. Analysis was performed blinded on ImageJ-win64. $n = 5$ biologically independent samples, mean ± sd. ****$P \leq 0.0001$, unpaired $t$-test. (C) Immunoblot of the nuclear fraction of HeLa-Cas9 incubated in 1% oxygen for 0 to 60 min. USP43 mixed KO population is shown in Fig. EV4A. (D) Immunoblot of the soluble (S) and chromatin fractions (Ch) obtained using a formaldehyde crosslinking chromatin extraction protocol. HeLa-Cas9 and USP43 mixed population KO cells were incubated in 21 or 1% oxygen for 6 h prior to lysis. (E) Quantification of HIF-1α enrichment within the nuclear fraction in HeLa control or USP43 mixed population KO cells. Cells were incubated in 1% oxygen for 0 to 60 min prior to lysis, and HIF-1α levels relative to a stable histone mark (H3K36me3) were measured by immunoblot. The representative immunoblot is shown in Fig. EV4C. $n = 3$ biologically independent samples, mean ± sd. Control vs. USP43 sgRNA 1% $O_2$ ***$P = 0.0005$, two-way ANOVA. (F) As for (D) but with HeLa control and USP43 overexpression (OE). Representative immunoblot is shown in Fig. EV4D. (G) As for (E) but with HeLa control and USP43 overexpression. $n = 3$ biologically independent samples, mean ± sd. Control vs. USP43 sgRNA 1% $O_2$ *$P = 0.0125$, two-way ANOVA. Source data are available online for this figure.

implicated. DUBs that counteract VHL may also only be uncovered at lower oxygen concentrations, as noted with USP33 (Zhang et al, 2022). Alternatively, counteracting VHL-mediated ubiquitination may not be required, as the oxygen-sensitivity of prolyl hydroxylation provides the regulation and rate-control within the pathway. In contrast, several DUBs can influence the activity of the HIF response, seemingly independently of VHL-mediated ubiquitination. USP43, USP52, JOSD1, UCHL5 and OTUB1 all reduced levels of the HIF-1α target CA9 in the secondary validation screen in two different non-cancer cell lines. OTUB1 has been previously implicated in hypoxia-mediated turnover of HIF-1α (Liu et al, 2022a). UCHL5 is a proteasome-associated DUB that has been previously implicated in HIF regulation through ubiquitin recycling (Kim et al, 2015). JOSD1 has not been linked to HIF signalling and warrants further investigation.

The involvement of USP43 in the HIF-1 hypoxia response highlights that coordinated activation HIF signalling involves multiple layers of regulation to refine the response. While prolyl hydroxylation and VHL-mediated ubiquitination are the essential core regulators of the HIF pathway, specific transcriptional responses are mediated downstream, either through the recruitment of co-activators (Batie et al, 2019; Batie et al, 2022; Chakraborty et al, 2019; Galbraith et al, 2013; Liu et al, 2022b; Ortmann et al, 2021), the local chromatin environment (Batie et al, 2022), or through HIF isoform interactions (Smythies et al, 2019). The association of USP43 with 14-3-3 to facilitate a HIF-1-specific response provides a further layer of complexity in how HIF signalling is controlled and helps explain how differential transcription programmes are initiated.

## Methods

### Reagents and tools table

| Reagent/Resource | Reference or source | Identifier or catalogue number |
|---|---|---|
| **Experimental Models** | | |
| HeLa (*H. sapiens*) | Gift from Lehner group, CITIID, University of Cambridge | Authenticated by Eurofins |
| HEK293T (*H. sapiens*) | ECACC | 12022001 |
| MCF7 (*H. sapiens*) | Gift from Adrian L. Harris laboratory, Department of Oncology, University of Oxford | |
| A549 (*H. sapiens*) | ECACC | 86012804 |
| RCC4 (*H. sapiens*) | Gift from Maxwell lab, Cambridge Institute for Medical Research | |
| 786-O (*H. sapiens*) | Gift from Maxwell lab, Cambridge Institute for Medical Research | |
| RPE-1 (*H. sapiens*) | Gift from Sinclair lab, University of Cambridge | |
| HKC-8 (*H. sapiens*) | Gift from Maxwell lab, Cambridge Institute for Medical Research | |
| **Recombinant DNA** | | |
| pKLV-U6gRNA-EF(BbsI)-PGKpuro2ABFP | Addgene | 62348 |
| pHRSIN-pSFFV-HA-pPGK-Puro/Hygro | Gift from Paul Lehner, University of Cambridge | |
| pHRSIN-pSFFV-pPGK-Puro/Hygro | Gift from Paul Lehner, University of Cambridge | |
| pMD.G (Lentiviral VSVG) | Gift from Paul Lehner, University of Cambridge | (Demaison et al, 2002) |
| pMD.GagPol (Lentiviral Gag/Pol) | Gift from Paul Lehner, University of Cambridge | (Demaison et al, 2002) |
| HIF-1α-ODDGFP reporter | Generated in Nathan lab | (Burr et al, 2016) |
| pCR4-TOPO USP43 | Source BioScience | IMAGE: 9052561 |
| pHRSIN-FLAG-NLS-Cas9-NLS-pGK-Hygro | Gift from Paul Lehner, University of Cambridge | |
| **Antibodies** | | |
| Mouse monoclonal anti-β actin | Sigma | A2228 |
| Mouse monoclonal anti-CA9 (M75) | Absolute Antibody | Ab00414-1.1 |
| Rabbit monoclonal anti-H3 | Cell Signalling | 4499 |

| Antibodies | | |
|---|---|---|
| Rabbit monoclonal anti-H3K4me3 | Cell Signalling | 9751 |
| Rabbit monoclonal anti-H3K36me3 | Cell Signalling | 4909 |
| Rat monoclonal anti-HA | Roche | 11867423001 |
| Mouse monoclonal anti-HIF-1α | BD Biosciences | 610959 |
| Rabbit monoclonal anti-HIF-1α | Cell Signalling | 36169 |
| Rabbit monoclonal anti-HIF-2α | Cell Signalling | 7096 |
| Rabbit monoclonal anti-HIF-1β | Cell Signalling | 5537 |
| Rabbit polyclonal anti-USP43 | Atlas AB | HPA027762 |
| Rabbit polyclonal anti-USP43 | Atlas AB | HPA023389 |
| Rabbit polyclonal anti-PHD2 | NovusBio | NB100-137 |
| Rabbit polyclonal anti-PHD3 | Thermo Fisher Scientific | A300-327A |
| Rabbit monoclonal anti-H2BK120Ub1 | Cell Signalling | 5546 |
| Rabbit monoclonal anti-H2B | Cell Signalling | 12364 |
| Rabbit polyclonal anti-Phospho-(Ser) 14-3-3 Binding Motif Antibody | Cell Signalling | 9601 |
| Rabbit IgG | Cell Signalling | 2729 S |
| Mouse IgG | Cell Signalling | 5415 S |
| Mouse monoclonal anti-alpha Tubulin | eBioscience | 14-4502 |
| Streptavidin ExtrAvidin-Peroxidase | E2886 | 17970-125 |
| Alexa-Fluor 488 Goat Anti-Mouse IgG | Thermo | A32723 |
| Alexa-Fluor 488 Goat Anti-Rabbit IgG | Thermo | A11034 |
| Alexa-Fluor 647 Goat Anti-Mouse IgG | Thermo | A21236 |
| Peroxidase-AffiniPure Goat Anti-Mouse IgG | Jackson | 115-035-146 |
| Peroxidase-AffiniPure Goat Anti-Rabbit IgG | Jackson | 111-035-045 |
| Peroxidase-AffiniPure Goat Anti-Rat IgG | Jackson | 112-035-167 |
| **CRISPR sgRNA oligonucleotide sequences** | **Sequence** | |
| PHD2 sgRNA | ATGCCGTGCTTGTTCATGCA | |
| HIF-1α sgRNA | CCTCACACGCAAATAGCTGA | |
| HIF1β sgRNA | CAGTCCTCCGTCTCCTCACC | |
| HIF-2α sgRNA | GCTGATTGCCAGTCGCATGA | |
| USP43 sgRNA 1 | CACCTGTTTCATGAACGCG | |
| USP43 sgRNA 2 | GGACACTGTGATGGCGACG | |
| USP43 sgRNA 3 | GCGCAAGGGGATAGGTAGG | |
| USP43 sgRNA Pfizer | GTCAGCAGCACAGCTGTACCT | |
| USP52 sgRNA 1 | CAAGTATATGGCCCGTGGG | |
| USP52 sgRNA 2 | GTTCCACCTTAAAAGTACG | |
| USP52 sgRNA 3 | CAGACTTGGGTTCAGGTGG | |
| OTUD5 sgRNA g1 | GATGTACAACCGTCCTGTGGAGG | |
| OTUD5 sgRNA g2 | GAATATCCACTATAATTCAGTGG | |
| OTUD5 sgRNA g3 | GAAGAATGCCATAAAAACATCGG | |
| 14-3-3 beta sgRNA 1 | TGAAGGCAGTCACAGAACAGGGG | |
| 14-3-3 beta sgRNA 2 | GTGCCAGACCAAGACGAATTGGG | |
| 14-3-3 epsilon sgRNA 1 | GAGCAGGCTGAGCGATACGACGG | |
| 14-3-3 epsilon sgRNA 2 | GGCTGCGGAGAACAGCCTAGTGG | |
| 14-3-3 gamma sgRNA 1 | AGAGCCAGGCCTAATCGGATGGG | |
| 14-3-3 gamma sgRNA 2 | AGCAACTGGTGCAGAAAGCCCGG | |
| 14-3-3 eta sgRNA 1 | CAGCATTGAGCAGAAAACCATGG | |
| 14-3-3 eta sgRNA 2 | CGCGCTCACCGCCTTCATAGCGG | |
| 14-3-3 tau sgRNA 1 | TACTGTCTTTGCAAGAGAGCAGG | |

| CRISPR sgRNA oligonucleotide sequences | Sequence |
|---|---|
| 14-3-3 tau sgRNA 2 | AGTATTGAACAAAAGACGGAAGG |
| 14-3-3 zeta sgRNA 1 | AGATATCTGCAATGATGTACTGG |
| 14-3-3 zeta sgRNA 2 | GCCGCTGGTGATGACAAGAAAGG |
| 14-3-3 sigma sgRNA 1 | AGCCCGGTCAGCCTACCAGGAGG |
| 14-3-3 sigma sgRNA 2 | AGAGCAGGCCGAACGCTATGAGG |
| USP43 sgRNA 4 | GAACTGAAAACATTCATCCAAGG |
| USP43 sgRNA 5 | GATCCAAAGGACCACCGCAGAGG |
| USP52 sgRNA 4 | GCTGTGGGTGGGGAGCCACGGGG |
| USP52 sgRNA 5 | GGAGAGTGTGAAAAGCAGAGTGG |
| USP33 sgRNA 1 | GCAGTGTCTGACTTGTGACAGGG |
| USP33 sgRNA 2 | GCGAAGCATATGCTCCACAAGGG |
| USP15 sgRNA 1 | TGAACCAGCGACTATCGACTAGG |
| USP15 sgRNA 2 | GTTGGAATAAACTTGTCAGCTGG |
| JOSD1 sgRNA 1 | GGACAGCAATGCCTTCACCCGGG |
| JOSD1 sgRNA 2 | AGGCTATGAAGCTGTTTGGTGGG |
| STAMBP sgRNA 1 | GTCGACACTGGAGAGAAACGCGG |
| STAMBP sgRNA 2 | TGAAAAGAAAGACACAGTAAAGG |
| UCHL5 sgRNA 1 | GTTCAGTAACACACTCACTATGG |
| UCHL5 sgRNA 2 | CCTCATGGAAAGCGACCCCGGGG |
| OTUB1 sgRNA 1 | AGGGAGGCTGGAGAGTCCCACGG |
| OTUB1 sgRNA 2 | TGGGCCCAGAGGATTCCCTGAGG |
| CYLD sgRNA 1 | AGGTCCTGGGGACACAATGCAGG |
| CYLD sgRNA 2 | ACTTCACAAAAGCAACATAGTGG |
| USP25 sgRNA 1 | AGAGAAGCCAAAGAACCCCATGG |
| USP25 sgRNA 2 | GAAATAGAAAATGACACCAGAGG |

| siRNA | Source | Sequence/ Identifier |
|---|---|---|
| SMARTpool for USP43 | Dharmacon | L-023019-00-0005 |
| SMARTpool for Set1B | Dharmacon | L-027025-01-0005 |
| USP43 | Eurofins | 5′-GAA GAU GGU UGC AGA GGA A-3′ |
| MISSION siRNA Universal Negative Control | Merck | SIC002 |
| Scrambled negative control | Eurofins | 5′-CAG UCG CGU UUG CGA CUG G-3′ |
| 14-3-3 PAN siRNA | (Dar et al, 2014) | 5′-AAG CTG GCC GAG CAG GCT GAG CGA TA-3′ |

| qPCR Primers | |
|---|---|
| Actin Forward | CTGGGAGTGGGTGGAGGC |
| Actin Reverse | TCAACTGGTCTCAAGTCAGTG |
| ANGPTL4 Forward | TCCGTACCCTTCTCCACTTG |
| ANGPTL4 Reverse | AGTACTGGCCGTTGAGGTTG |
| BAP1 Forward | GGAGGTAGAGAAGAGGAAGAA |
| BAP1 Reverse | GAGCCAGCATGGAGATAAAG |
| CA9 Forward | GCCGCCTTTCTGGAGGA |
| CA9 Reverse | TCTTCCAAGCGAGACAGCAA |
| GLUT1 Forward | CCAGGGTAGCTGCTGGAGC |
| GLUT1 Reverse | TGGCATGGCGGGTTGT |
| HIF-1α Forward | CCAGTTACGTTCCTTCGATCAGT |
| HIF-1α Reverse | TTTGAGGACTTGACTTGCGCTTTCA |
| PAI-1 Forward | ACCGCAACGTGGTTTTCTCA |
| PAI-1 Reverse | TTGAATCCCATAGCTGCTTGAAT |
| PHD3 Forward | TCCTGCGGATATTTCCAGAGG |

| qPCR Primers | |
|---|---|
| PHD3 Reverse | GGTTCCTACGATCTGACCAGAA |
| PDK-1 Forward | ATGTACCATCCCATCTCTATCAC |
| PDK-1 Reverse | GGTCACTCATCTTCACAGTC |
| VEGF Forward | TACCTCCACCATGCCAAGTG |
| VEGF Reverse | ATGATTCTGCCCTCCTCCTTC |
| BAP1 Forward | GGAGGTAGAGAAGAGGAAGAA |
| BAP1 Reverse | GAGCCAGCATGGAGATAAAG |
| 14-3-3 beta Forward | GGCAAAGAGTACCGTGAGAAG |
| 14-3-3 beta Reverse | CTGGTTGTGTAGCATTGGGAATA |
| 14-3-3 epsilon Forward | GATTCGGGAATATCGGCAAATGG |
| 14-3-3 epsilon Reverse | GCTGGAATGAGGTGTTTGTCC |
| 14-3-3 gamma Forward | AGCCACTGTCGAATGAGGAAC |
| 14-3-3 gamma Reverse | CTGCTCAATGCTACTGATGACC |
| 14-3-3 eta Forward | GACATGGCCTCCGCTATGAAG |
| 14-3-3 eta Reverse | ATGCTGCTAATGACCCTCCAG |
| 14-3-3 tau Forward | AGGGTCATCTCTAGCATCGAG |
| 14-3-3 tau Reverse | CCACTTTCTCCCGATAGTCCTT |
| 14-3-3 zeta Forward | CCTGCATGAAGTCTGTAACTGAG |
| 14-3-3 zeta Reverse | GACCTACGGGCTCCTACAACA |
| 14-3-3 sigma Forward | TGACGACAAGAAGCGCATCAT |
| 14-3-3 sigma Reverse | GTAGTGGAAGACGGAAAAGTTCA |
| **ChIP qPCR primers** | |
| CA9-2 Forward | CTTCTGGTGCCTGTCCATC |
| CA9-2 Reverse | ATCCTCCTCTCTGGGTGAAT |
| CA9-1 Forward | GACAAACCTGTGAGACTTTGGCTCC |
| CA9-1 Reverse | AGTGACAGCAGCAGTTGCACAGTG |
| PHD3 Forward | TGCGGGATCAAGATTCTATTC |
| PHD3 Reverse | GCCTTTCAGTGGCTTCTT |
| VEGF Forward | CCTTTGGGTTTTGCCAGA |
| VEGF Reverse | CCAAGTTTGTGGAGCTGA |
| **USP43 cloning primers** | |
| pHRSIN-pSFFV-HA-pPGK-Puro/Hygro Forward/ Reverse | 5′-GCTTATCCTTACGACGTGCCTGACTACGCCGGATCCGACCTGGGCCCCGGGGACGCGGCAGGAG-3′ 5′-CTTGCATGCCTGCAGGTCGACTCTAGAGTCGCGGCCGCTCAAAAGCTGGACTCCGGTAAGGTTTTCTTTCG-3′ |
| pHRSIN-pSFFV-pPGK-Puro/Hygro Forward/ Reverse | 5′-CTCTATAAAAGAGCTCACAACCCCTCACTCGGCGCGCCAGTCCTCCGACAGACTGAGTC-3′ 5′-CTTGCATGCCTGCAGGTCGACTCTAGAGTCGCGGCCGCTCAAAAGCTGGACTCCGGTAAGGTTTTCTTTCG-3′ |
| PAM SDM | CATCAGCAGCACAGCTGTACTTTGGATGAATGTTTCAGTTC |
| USP43<sup>C11A</sup> SDM | GAAGAACCACGGCAACACCGCTTTCATGAACGCGGTGGTGCAG |
| USP43<sup>H668R</sup> SDM | GGCAACCTGCAAGGTGGGCGTTACACAGCCTACTGCCGGAAC |
| **DUB screening primers** | |
| PCR1 Forward | AGGGCCTATTTCCCATGATTCCTT |
| PCR1 Reverse | TCAAAAAAGCACCGACTCGG |
| PCR2 Forward | AATGATACGGCGACCACCGAGATCTACACTCTCTTGTGGAAAGGACGAAACACCG |
| PCR2 Reverse | CAAGCAGAAGACGGCATACGAGAT NNNNNNN GTGACTGGAGTTCAGACGTGTGCTCTTCCGA TCTCCATTTGTCACGTCCTGCACG |
| PCR2 Index2 | CAAGCAGAAGACGGCATACGAGAT TACATCG GTGACTGGAGTTCAGACGTGTGCTCTTCCGA TCTCCATTTGTCACGTCCTGCACG |
| PCR2 Index4 | CAAGCAGAAGACGGCATACGAGAT TTGGTCA GTGACTGGAGTTCAGACGTGTGCTCTTCCGAT CTCCATTTGTCACGTCCTGCACG |
| PCR2 Index7 | CAAGCAGAAGACGGCATACGAGAT TGATCTG GTGACTGGAGTTCAGACGTGTGCTCTTCCGA TCTCCATTTGTCACGTCCTGCACG |

| DUB screening primers | | | |
|---|---|---|---|
| PCR2 Index16 | CAAGCAGAAGACGGCATACGAGAT GGGACGG GTGACTGGAGTTCAGACGTGTGCTCTTCCGATC TCCATTTGTCACGTCCTGCACG | | |
| Sequencing primer | ACACTCTCTTGTGGAAAGGACGAAACACCG | | |
| **Chemicals, enzymes and other reagents** | | **Source** | **Catalogue number** |
| Dimethyloxalylglycine (DMOG) | | Cayman Chemical | 71210 |
| MG132 | | Sigma | 474790 |
| Cyclohexamide | | Sigma | 01810 |
| Roxadustat (FG-4592) | | Cayman Chemicals | 15294 |
| Biotin-ANP-Ub-PA | | UbiQ | UbiQ-077 |
| AF1 + DAPI | | CitiFluor | 17970-125 |
| Lambda Phosphatase | | NEB | P0753 |
| Calyculin A | | CST | 9902 |
| Fugene | | Promega | E5911 |
| HEPES | | Sigma | H0887 |
| Gentra Puregene Core kit | | Qiagen | 158388 |
| Benzonase | | Sigma | E1014 |
| Denarase | | c-LEcta | |
| Pierce ECL Western Blotting Substrate | | Thermo Scientific | 32106 |
| Supersignal West Pico Plus Chemiluminescent substrate | | Thermo Scientific | 34580 |
| RNeasy Plus Minikit | | Qiagen | 74134 |
| PureLink RNA Mini Kit | | Thermo Scientific | 12183025 |
| Protoscript II Reverse Transcriptase | | NEB | M0368X |
| Protein G magnetic beads | | Thermo Scientific | 88848 |
| EDTA-free protease inhibitor cocktail tablet | | Roche | 11873580001 |
| XF DMEM, pH 7.4 | | Agilent Technologies | 103575-100 |
| Sodium pyruvate | | Life Technologies | 103578-100 |
| L-glutamine | | Life Technologies | 103579-100 |
| Glucose | | Sigma Aldrich | 103577-100 |
| rotenone | | Sigma Aldrich | R8875 |
| antimycin A | | Sigma Aldrich | 1397-94-0 |
| 2-deoxyglucose | | Alfa Aesar | 154176 |
| Hoesht | | Thermo Scientific | 33342 |
| proteinase K | | Thermo Scientific | EO0491 |
| RNase A (Thermo Scientific | | Thermo Scientific | EN0531 |
| DNA minielute kit | | QIagen | 28004 |
| anti-HA magnetic beads | | Thermo Scientific | 88836 |
| Lipofectamine RNAi MAX | | Thermo Scientific | 13778-075 |
| **Software** | | | |
| GraphPad Prism v.8 | https://www.graphpad.com | | |
| ImageJ | https://imagej.nih.gov/ij/index.html | | |
| ImageJ Fiji | https://fiji.sc/ | | |
| Snakemake | https://doi.org/10.12688/f1000research.29032.2 | | |
| FastQC | https://www.bioinformatics.babraham.ac.uk/projects/fastqc/ | | |
| MultiQC | https://doi.org/10.1093/bioinformatics/btw354 | | |
| Cutadapt | https://doi.org/10.14806/ej.17.1.200 | | |
| HISAT2 | https://doi.org/10.1038/s41587-019-0201-4 | | |
| MAGeCK | https://doi.org/10.1186/s13059-014-0554-4 | | |
| Tidyverse | https://doi.org/10.21105/joss.01686 | | |
| Cowplot | https://github.com/wilkelab/cowplot | | |

| Software | |
|---|---|
| WAVE v.2.6.1 | https://www.agilent.com/en/product/cell-analysis/real-time-cell-metabolic-analysis/xf-software/seahorse-wave-controller-software-2-6-1-740904 |
| TrimGalore | https://doi.org/10.5281/zenodo.7598955 |
| Salmon | https://doi.org/10.1038/nmeth.4197 |
| DESeq2 | https://doi.org/10.1186/s13059-014-0550-8 |
| **Other** | |
| Whitley H35 Hypoxystation | Don Whitley Scientific |
| Fortessa | e6 BD Fortessa |
| Influx cell sorter | 6 BD Influx |
| ABI 7900HT Real-Time PCR system | Applied Biotechnology or Quantstudio 7, Thermo Scientific |
| HiSeq | Illumina |
| Bioruptor tubes for DNA Shearing | Diagenode |
| Biorupter | Diagenode |
| Seahorse XF analyzer | Agilent Technologies |
| Seahorse XF FluxPak consumables | Agilent Technologies |
| CLARIOstar Plate Reader | BMG Labtech |
| LSM 980 with Airyscan confocal microscope | Zeiss |
| MiniSeq | Illumina |

## Cell culture and reagents

HeLa, HEK293T, MCF7 and A549 cells were maintained in DMEM (Sigma Aldrich) and supplemented with 10% FCS; RCC4 and 786-0 cells were maintained in RPMI-1640 (Sigma) supplemented with 10% FCS. Hypoxic cell culture was performed in a Whitley H35 Hypoxystation (Don Whitley Scientific) at 37 °C, 5% $CO_2$, 1% $O_2$ and 94% $N_2$. Cells were confirmed mycoplasma negative (Lonza, MycoAltert), and authenticated by short tandem repeat profiling (Eurofins Genomics). In each instance of drug treatments, controls were treated with their vehicle controls (DMSO).

## Plasmids

USP43 constructs were generated from pCR4-TOPO USP43 (Source BioScience, IMAGE:9052561) and cloned into the pHRSIN-pSFFV backbone with puromycin and hygromycin resistance using NEBuilder HiFi (NEB).

## Lentiviral production and transduction

Lentivirus was produced by transfection of HEK293T cells with Fugene (Promega) at 70–80% confluency in six-well plates, with the appropriate pHRSIN vector and the packaging vectors pCMVR8.91 (gag/pol) and pMD.G (VSVG). Viral supernatant was harvested at 48–72 h, filtered (0.22-μm filter), and stored at −80 °C. For transduction, cells were seeded on 24-well plates in 500 μl medium, and 500 μl viral supernatant was added. Antibiotic selection was applied for 48 h.

## Flow cytometry

A total of $2 \times 10^5$ cells per sample were washed in ice-cold PBS in 5-ml FACS tubes and resuspended in 500 μl PBS/formaldehyde before analysis on a BD Fortessa (GFP). For cell-surface staining, cells were washed in ice-cold PBS, incubated at 4 °C for 30 min with the primary antibody, washed with PBS and incubated with the appropriate secondary antibody at 4 °C for 30 min.

## DUB library CRISPR–Cas9 forward genetic screen

The DUB sgRNA library was generated as a subpool of our 'ubiquitome library' (Menzies et al, 2018). Clonal HeLa HRE-ODDGFP cells were transduced with *Streptococcus pyrogenes* Cas9 (pHRSIN-FLAG-NLS-Cas9-NLS-pGK-Hygro) and selected for Cas9 expression using hygromycin. A total of $6 \times 10^6$ HeLa HRE-ODDGFP cells were transduced with pooled sgRNA virus (multiplicity of infection of around 0.3), establishing at least 500-fold sgRNA coverage, which was maintained throughout the experiments. After 30 h, cells were treated with puromycin 1 μg ml⁻¹ for 5 days. The library was pooled before any selection event. After 8 days, we performed FACS by harvesting and sorting $10^8$ cells after 24 h of 1% $O_2$, washing the cells in PBS and resuspending in PBS with 2% FCS and 10 mM HEPES. Cells were sorted using an Influx cell sorter and kept on ice through this process to maintain the stability of the reporter. GFP^Low cells were chosen in a gate set at 1 $\log_{10}$ unit below the mode of the 1% $O_2$ treated control population. Representative gating strategies are shown (Appendix Fig. S5). Cells were then harvested for DNA extraction. Genomic DNA was extracted using a Gentra Puregene Core kit. Lentiviral sgRNA inserts were amplified in a two-step PCR (with Illumina adaptors added on the second PCR).

A Snakemake pipeline (https://doi.org/10.5281/zenodo.10286661) was used for bioinformatic analysis: after investigating the quality of the raw data using FastQC and MultiQC, reads were quality trimmed to 20 bp using Cutadapt. Trimmed reads were then aligned to all sgRNA sequences using HISAT2 (>90% alignment rates for all samples). To determine the changes in normalised read counts between samples,

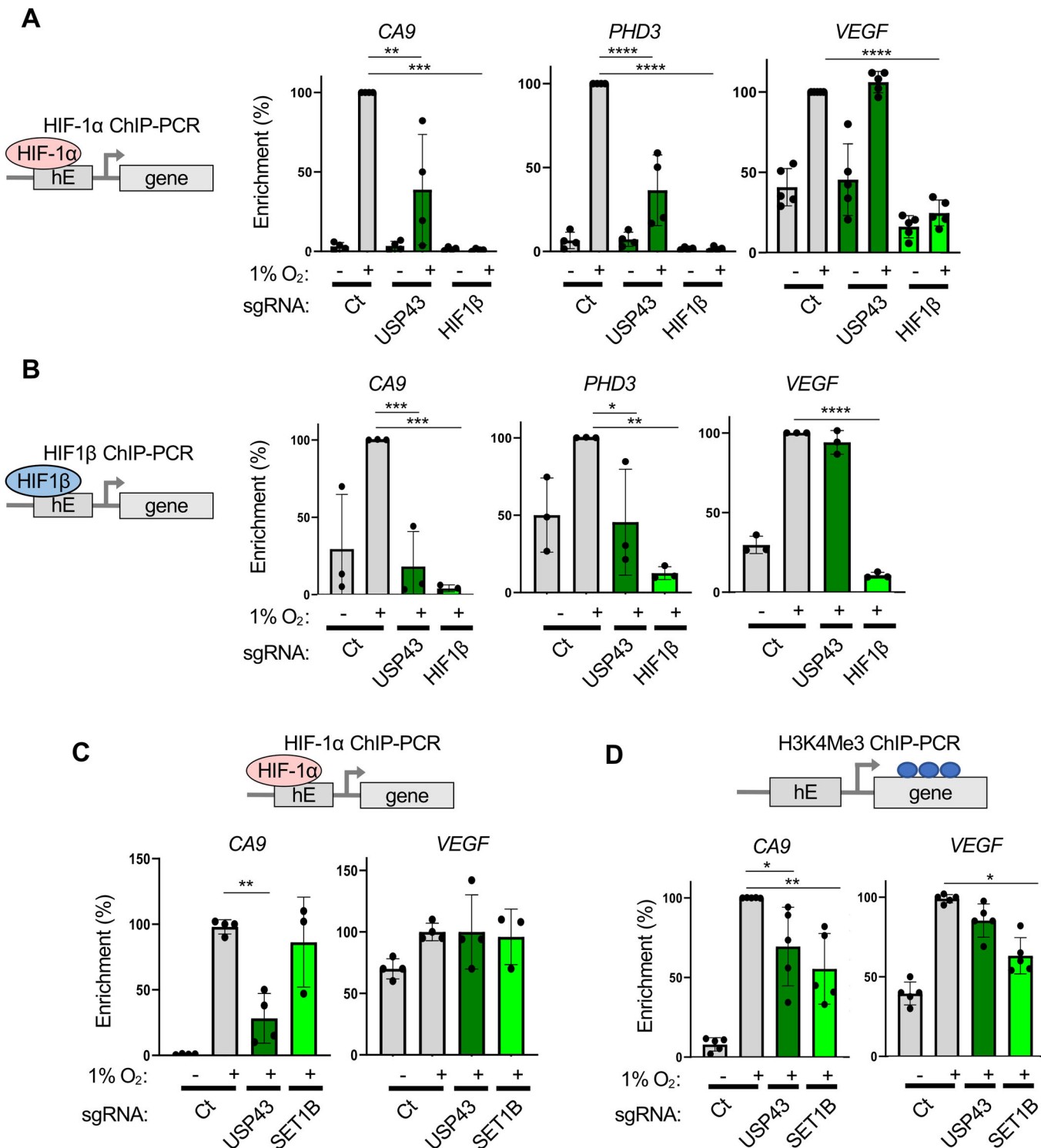

**Figure 6. USP43 KO reduces HIF-1 binding to chromatin at HREs.**

(A) HIF-1α ChIP-PCR for HREs of *CA9, PHD3* and *VEGF* in control, HIF1β or USP43 depleted HeLa cells treated with 21 or 1% oxygen for 6 h. $n = 4$ biologically independent samples, mean ± sd. *CA9*: control vs. HIF1β sgRNA ***$P = 0.0001$, control vs. USP43 sgRNA **$P = 0.0036$. *PHD3*: control vs. HIF1β sgRNA ****$P < 0.0001$, control vs. USP43 sgRNA ****$P < 0.0001$. *VEGF*: control vs. HIF1β sgRNA ****$P < 0.0001$. One-way ANOVA. (B) as (A) but HIF1β binding to specific HREs. $n = 3$ biologically independent samples, mean ± sd. *CA9*: control vs. HIF1β sgRNA ***$P = 0.0002$, control vs. USP43 sgRNA ***$P = 0.0005$. *PHD3*: control vs. HIF1β sgRNA **$P = 0.003$, control vs. USP43 sgRNA *$P = 0.0266$. *VEGF*: control vs. HIF1β sgRNA ****$P < 0.0001$. One-way ANOVA. (C) HIF-1α ChIP-PCR at *CA9* or *VEGF* HREs in HeLa control, USP43 depleted ($n = 4$ biological repeats, mean ± sd) or SET1B depleted ($n = 3$ biological repeats, mean ± sd) HeLa cells. Cells were incubated in 21 or 1% oxygen for 6 h prior to lysis. *CA9*: control vs. USP43 sgRNA, **$P = 0.0031$, one-way ANOVA. (D) As for (C) but immunoprecipitated for H3K4me3 at the HREs of *CA9* or *VEGF* ($n = 5$ biological repeats). *CA9*: control vs. USP43 sgRNA *$P = 0.0444$, control vs. Set1B sgRNA **$P = 0.0056$. *VEGF*: control vs. Set1B sgRNA ****$P < 0.0001$. One-way ANOVA. Source data are available online for this figure.

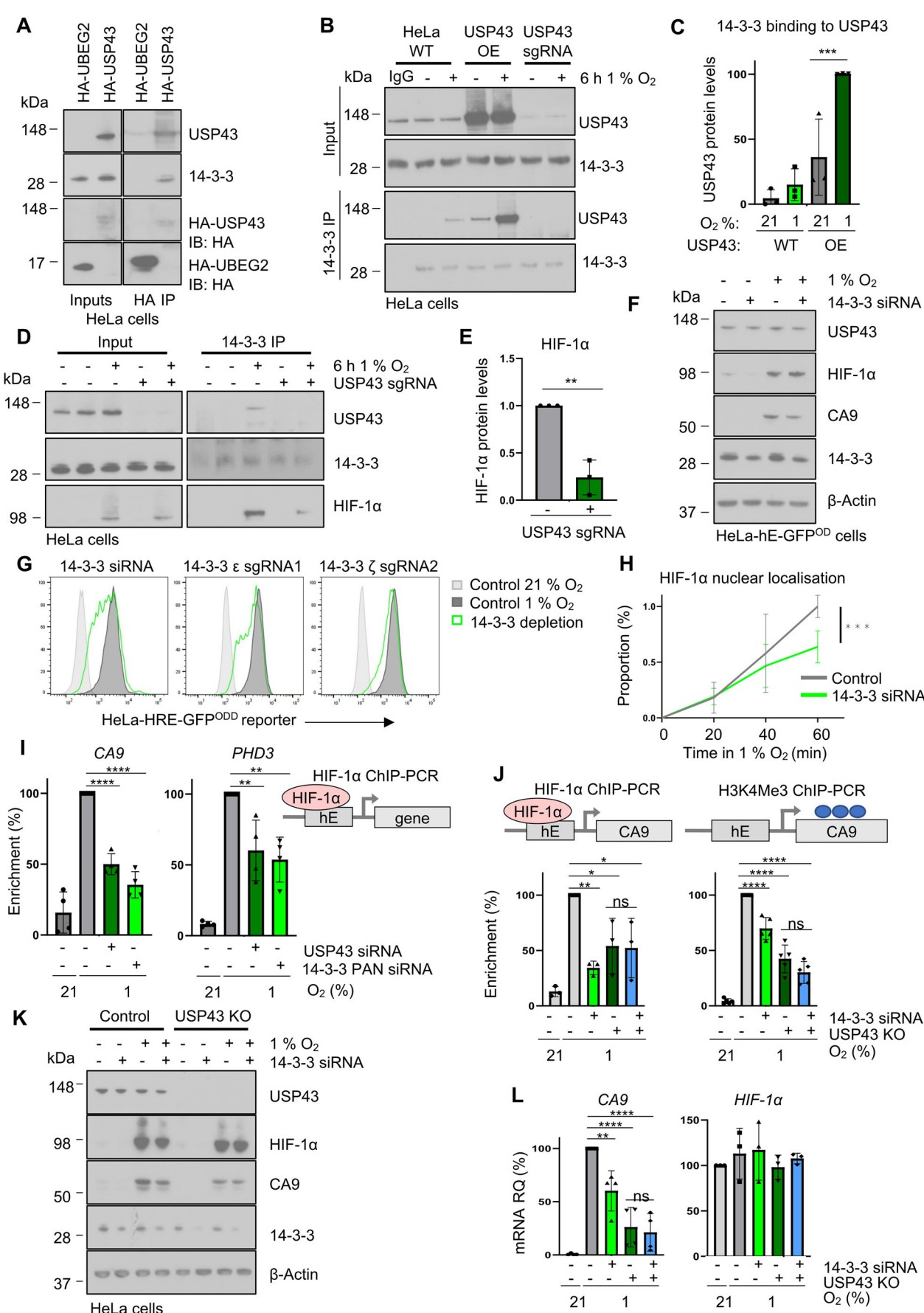

◄ **Figure 7. USP43 associates with 14-3-3 to regulate HIF-1 signalling.**

(A) HeLa cells stably expressing HA-USP43 or HA-UBEG2 were immunoprecipitated and immunoblotted using a pan 14-3-3 antibody (recognises all isoforms). (B, C) Control, USP43 depleted (sgRNA) or overexpressing HeLa cells were incubated at 21 or 1% oxygen for 6 h prior to lysis. Endogenous 14-3-3 proteins were immunoprecipitated and samples were immunoblotted for USP43 (B). Quantification of USP43 associated with immunoprecipitated 14-3-3 (C). USP43 protein levels normalised to USP43 input. $n = 3$ biological replicates, mean ± sd. ***$P = 0.0004$, one-way ANOVA. (D, E) Control or USP43 depleted HeLa cells were incubated in 21 or 1% oxygen for 6 h prior to lysis. Endogenous 14-3-3 was immunoprecipitated and immunoblotted for USP43 and HIF-1α (D). Quantification of HIF-1α association with 14-3-3 proteins (relative to HIF-1α input protein level) (E). $n = 3$ biological replicates, mean ± sd. **$P = 0.002$, unpaired *t*-test. (F) Immunoblot of HeLa HRE-$^{ODD}$GFP reporter and 14-3-3 siRNA treated cells after 16 h of 21 or 1% oxygen treatment. Immunoblot is representative of three biological replicates. (G) HeLa HRE-$^{ODD}$GFP reporter cells treated with 14-3-3 pan siRNA, 14-3-3 epsilon sgRNA or 14-3-3 zeta sgRNA were analysed by flow cytometry. Control levels of the reporter in 21% (light grey) or 1% (dark grey) oxygen are shown. (H) Quantification of HIF-1α enrichment within the nuclear fraction in HeLa control or following 14-3-3 siRNA-mediated depletion. Cells were incubated in 1% oxygen for 0 to 60 min prior to lysis, and HIF-1α levels relative to a stable histone mark (H3K36me3) were measured by immunoblot. The representative immunoblot is shown in Fig. EV5K. $n = 3$ biologically independent samples, mean ± sd. ***$P = 0.0006$, two-way ANOVA. (I) HIF-1α ChIP-PCR for HREs of *CA9* or *PHD3* in control, USP43 or 14-3-3- siRNA-depleted HeLa cells treated with 21 or 1% oxygen for 6 h. $n = 4$ biologically independent samples, mean ± sd. *CA9*: control vs USP43 siRNA ****$P < 0.0001$, control vs. 14-3-3 siRNA ****$P < 0.0001$. *PHD3*: control vs USP43 siRNA **$P = 0.0095$, control vs. 14-3-3 siRNA **$P = 0.0039$. One-way ANOVA. (J–L) Control or USP43 null HeLa cells, with or without 14-3-3 depletion, were incubated in 21 or 1% oxygen for 6 h. Samples were analysed by ChIP-PCR for HIF-1α ($n = 3$ biological repeats) or H3K4me3 ($n = 5$ biological repeats) at the HRE of *CA9* (J), immunoblot of HIF-1α, 14-3-3, CA9 and USP43 (representative of three biological replicates) (K) or qPCR for *CA9* ($n = 4$ biological repeats) and *HIF-1α* ($n = 3$ biological repeats) mRNA expression after 16 h incubation in 21 or 1% oxygen (L). (J) *HIF-1α*: control vs. USP43 sgRNA **$P = 0.0024$, control vs. 14-3-3 siRNA *$P = 0.023$, control vs. USP43 sgRNA + 14-3-3 siRNA *$P = 0.0186$. H3K4me3: control vs. USP43 sgRNA ****$P < 0.0001$, control vs. 14-3-3 siRNA ****$P < 0.0001$, control vs. USP43 sgRNA + 14-3-3 siRNA ****$P < 0.0001$. Mean ± sd, one-way ANOVA. (L) Control vs. 14-3-3 siRNA **$P = 0.0043$, control vs. USP43 sgRNA ****$P < 0.0001$, control vs. USP43 sgRNA + 14-3-3 siRNA ****$P < 0.0001$, mean ± sd, one-way ANOVA. Source data are available online for this figure.

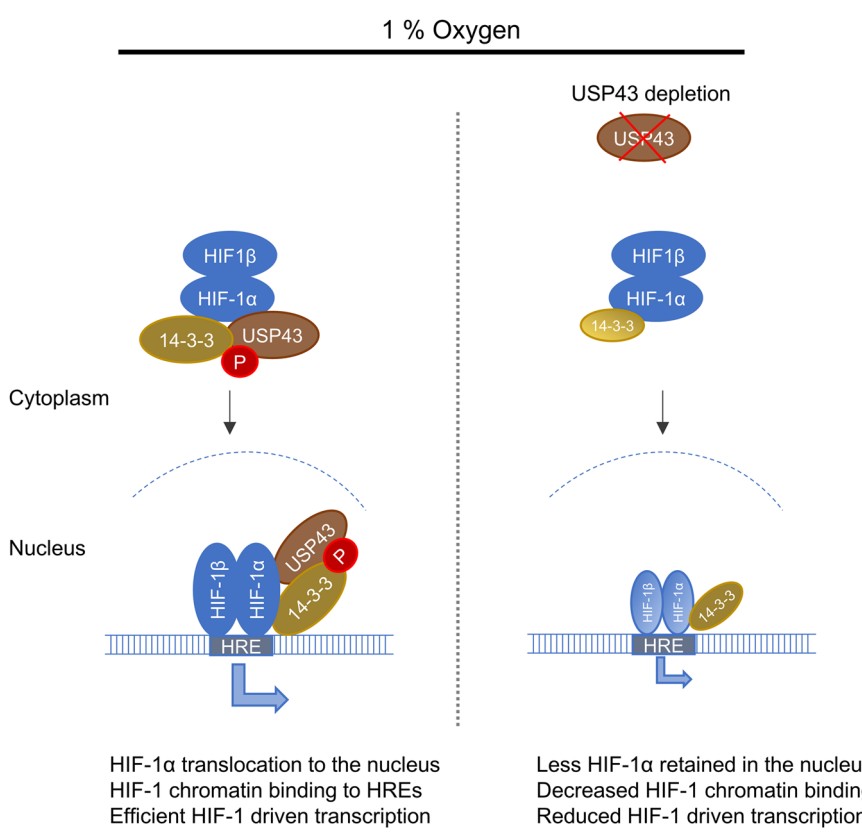

**Figure 8. USP43 mediated regulation of a HIF-1 transcriptional response.**

Schematic model of the role of USP43 in the HIF response. In hypoxia, USP43 associates with the stabilised HIF-1α in a phosphorylation and 14-3-3 dependent manner. This USP43 association promotes HIF-1 accumulation in the nucleus, and is required for the efficiency of HIF-1 to HREs on chromatin to activate gene transcription.

MAGeCK was used. The analysis presented compares DNA extracted following the sort to an unsorted DNA library taken at the same time point (Dataset EV1). All plots showing MAGeCK output were generated with a custom script using the R packages Tidyverse and Cowplot. All scripts used to analyse data and generate plots are publicly available (https://doi.org/10.5281/zenodo.10229339).

## CRISPR–Cas9 targeted deletions

Gene-specific CRISPR sgRNA sequences were taken from the Vienna library (www.vbc-score.org) or for USP43, a gift from Pfizer, with 5′ CACC and 3′CAAA overhangs, respectively. SgRNAs were ligated into the pKLV-U6gRNA(BbsI)-PGKpuro2ABFP vector and lentivirus was produced as described. Transduced cells were selected with puromycin, and were generally cultured for 9–10 days before subsequent experiments to allow sufficient time for the depletion of the target protein. Mixed KO populations were used (denoted by sgRNA) unless otherwise specified. USP43 null clones were also generated from the sgRNA-targeted populations (denoted by KO). Clones were isolated by serial dilution or FACS. USP43 loss was confirmed by immunoblot.

## Immunoblotting

Cells were lysed in a 1xSDS lysis buffer (1% SDS, 50 mM Tris pH 6.8, 10% glycerol, 0.01% (w/v) bromophenol blue, 0.1 M DTT, and 1:500 Benzonase or Denarase for 10 min before heating at 90 °C for 5 min. Proteins were separated by SDS–PAGE, transferred to PVDF (polyvinylidene difluoride) membranes, probed with appropriate primary and secondary antibodies and developed using enhanced chemiluminescent or Supersignal West Pico Plus Chemiluminescent substrate.

## Quantitative real-time PCR (qRT-PCR)

Total RNA was extracted using the RNeasy Plus Minikit or PureLink RNA Mini Kit following the manufacturer's instructions and then reverse transcribed using Protoscript II Reverse Transcriptase (NEB). Template cDNA (20 ng) was amplified using the ABI 7900HT Real-Time PCR system (Applied Biotechnology or Quantstudio 7). Transcript levels of genes were normalised to a reference index of a housekeeping gene (β-actin).

## RNA sequencing analysis

Total RNA was extracted from HeLa cells using the RNeasy Plus Minikit. Library preparation and sequencing (HiSeq, Illumina) were undertaken by Genewiz. A Snakemake pipeline was used to analyse the data (https://doi.org/10.5281/zenodo.10139567): after investigating the quality of the raw data using FastQC and MultiQC, sequence reads were trimmed to remove adaptor sequences and nucleotides with poor quality using TrimGalore. Transcript quantification was performed with Salmon against the Gencode Homo sapiens reference transcriptome/genome (build 44) using a decoy-aware transcriptome index for accurate mapping. A mapping rate of >90% was observed for all samples. DESeq2 was used for differential transcript analysis. Genes with adjusted $P$ values <0.01 and $\log_2$(fold change) >0.5 were called differentially expressed genes for each comparison (Dataset EV2). The volcano plots, PCA plot and sample distance heat map were generated with a custom R script using the Cowplot, DESeq2 and Tidyverse R packages. All scripts used to analyse data and generate plots are publicly available (https://doi.org/10.5281/zenodo.10229339).

## Immunoprecipitation

HeLa cells were lysed in 1% Triton PBS or RIPA (25 mM Tris•HCl pH 7.6, 150 mM NaCl, 1%, NP-40, 1% sodium deoxycholate, 0.1% SDS), with 1× Roche complete EDTA-free protease inhibitor cocktail and 1:1000 Denarase for 30 min at 4 °C. Lysates were centrifuged at 14,000 rpm for 10 min, supernatants collected and precleared with Protein G magnetic beads for 2 h at 4 °C. Supernatants were then incubated with primary antibody either overnight (rotation at 4 °C) or with protein G magnetic beads pre-bound to the primary antibody for an hour in 1xPBS with 0.2% Tween (rotation at 4 °C). Protein G magnetic beads were then added for 2 h, and samples were then washed three times in 1xPBS with 1% Triton and once in 1xPBS. Bound proteins were eluted in 2× SDS loading buffer, separated by SDS–PAGE and immunoblotted.

For phosphor-Serine (14-3-3 motif) immunoprecipitation and subsequent immunoblotting, 1xPBS was substituted by 1xTBS buffer (50 mM Tris-Cl, pH 8, 150 mM NaCl) in all the wash steps.

## Subcellular fractionation

High salt fractionation
About $10 \times 10^6$ HeLa cells were washed in PBS, lysed in Buffer A (10 mM HEPES, 1.5 mM MgCl$_2$, 10 mM KCl, 0.5 mM DTT and EDTA-free protease inhibitor cocktail tablet, Roche) and incubated with rotation at 4 °C for 10 min. The supernatant containing cytosolic fractions were collected by centrifugation ($1400 \times g$ for 4 min at 4 °C). The pellets were again washed with Buffer A and collected by centrifugation ($1400 \times g$ for 4 min at 4 °C) to reduce cytosolic fractions leakage to other fractions. The nuclear pellet was resuspended in Buffer B (20 mM HEPES, 1.5 mM MgCl$_2$, 300 mM NaCl, 0.5 mM DTT, 25% glycerol, 0.2 mM EDTA and EDTA-free protease inhibitor cocktail tablet) for 10 min on ice to separate nucleoplasmic and chromatin fractions. Samples were centrifuged at $1700 \times g$ for 4 min at 4 °C, separating the soluble nucleoplasm from the insoluble chromatin fraction. The chromatin fraction was solubilized in a 2× SDS loading buffer containing 1:500 Benzonase or 1:200 Denarase.

## Chromatin fractionation

A cross-liking chromatin enrichment approach was used (Kustatscher et al, 2014) with some modifications. About $20 \times 10^6$ HeLa cells were washed and scraped in cold PBS. Cells were pelleted by centrifugation at 1800×g for 10 min at 4 °C. The pellets were resuspended in 1 ml of cold lysis buffer (15 mM TRIS pH 7.5, 60 mM KCl, 15 mM NaCl, 1 mM CaCl$_2$, 250 mM Sucrose, 0.3% NP-0.3, Protease Inhibitor Cocktail) and incubated for 30 min at 4 °C with rotation. The cytoplasmic extract was separated by centrifugation at 600×g for 5 min at 4 °C and lysed in 2xSDS for immunoblotting. The nuclear fraction was washed twice with 1 ml of wash buffer (15 mM TRIS pH 7.5, 60 mM KCl, 15 mM NaCl, 1 mM CaCl$_2$, 250 mM Sucrose) spinning at 600×g, 5 min, 4 °C. The pellet was fixed in 500 µL 1% formaldehyde in PBS and incubated at 37 °C for 10 min, then neutralised with 500 µL 500 mM glycine in PBS at RT for 5 min (600×g, 5 min). The RNA was removed from the fraction by

incubating with 200 μg/mL RNase A in 500 μL of wash buffer for 15 min at 37 °C in low adhesion 2 ml Eppendorf tube (600×*g*, 5 min). The pelleted fraction was resuspended in 500 μL of SDS buffer (5% SDS in 25 mM TRIS pH 7.5) and 1.5 ml of urea buffer (8 M urea in 25 mM TRIS pH 7.5), mixed by inverting and centrifuged at 16,000×*g* for 30 min at RT. The SDS and urea wash was repeated twice. The translucent pellet was washed in 1 ml of SDS buffer (16,000×*g* for 5 min). The chromatin fraction in 100 μL of SDS buffer was sonicated using Bioruptor 0.5 ml Microtubes for DNA Shearing (ten times, 30 s on/off in a Biorupter). Crosslinked were reversed by heating at 95 °C for 45 min, and the protein samples dissolved in 100 μL of 2xSDS lysis buffer with 1:200 Denarase for immunoblotting.

## Bioenergetic analyses

Extracellular acidification rates and oxygen consumption rates were measured using a Seahorse XF analyser to calculate basal glycolysis. We performed the Real-Time ATP rate assay according to the manufacturer's instructions using Seahorse XF FluxPak consumables (Agilent Technologies). Briefly, A549 cells were depleted of HIF-1β or USP43 by sgRNA. Cells were plated in FluxPak 96-well plates in DMEM plus 10% FBS at a cell density of $2 \times 10^4$ with or without 100 μM Roxadustat and incubated at 37 °C for 24 h. The medium was replaced with XF DMEM, pH 7.4 (Agilent Technologies) supplemented with 1 mM sodium pyruvate, 2 mM L-glutamine, 10 mM glucose and incubated at 37 °C in a non-$CO_2$ incubator for 1 h before replacing with the same medium before measuring in the analyser. Programme settings: mix 3 min, measure 3 min × 3; inject 1.5 uM oligomycin; mix 3 min measure 3 min × 3; inject 0.5 μM rotenone, 0.5 μM antimycin A and 2.5 uM Hoechst; mix 3 min, measure 3 min × 3. Following the assay cells were quantified using the CLARIOstar Plate Reader at 355-20/455-30 nm (Hoechst detection). Data was analysed with GraphPad Prism v.8.

## Chromatin immunoprecipitation PCR (ChIP-PCR)

HeLa cells were grown on 15-cm dishes to $15 \times 10^6$ density, and then treated with 1% formaldehyde for 10 min to crosslink proteins to chromatin. The reaction was quenched with glycine (0.125 M, 10 min at room temperature). Cells were washed in ice-cold PBS twice, scraped in tubes, and centrifuged at 800 rpm for 10 min before lysis in 500 μl of ChIP lysis buffer (50 mM Tris-HCl (pH 8.1), 1% SDS, 10 mM EDTA, Complete Mini EDTA-free protease inhibitor). Samples were incubated on ice for 10 min and diluted 1:1 with ChIP dilution buffer (20 mM Tris-HCl (pH 8.1), 1% (v/v) Triton X-100, 2 mM EDTA and 150 mM NaCl). Samples were then sonicated for 20 cycles of 30 s on and 30 s off in a Biorupter, followed by centrifugation for 10 min at 13,000 rpm at 4 °C. Supernatants were collected and 20 μl stored at −20 °C as the input sample. A 200 μl aliquot of the remaining sample was diluted with ChIP dilution buffer to 1 ml, and precleared using 25 μl Protein G magnetic beads (4 °C, 2 h, rotating); 1 ml of sample was immunoprecipitated with the appropriate primary antibody (4 °C, overnight, rotating). Protein G magnetic beads (25 μl) were added and incubated for a further 2 h at 4 °C. The beads were washed sequentially for 5 min each with wash buffer 1 (20 mM Tris-HCl (pH 8.1), 0.1% (w/v) SDS, 1% (v/v) Triton X-100, 2 mM EDTA, 150 mM NaCl), wash buffer 2 (Wash Buffer 1 with 500 mM NaCl), wash buffer 3 (10 mM Tris-HCl (pH 8.1), 0.25 M LiCl, 7 1% (v/v)

NP-40, 1% (w/v) Na-deoxycholate and 1 mM EDTA), and twice with TE buffer (10 mM Tris-HCl (pH 8.0) and 1 mM EDTA). Bound complexes were eluted with 120 μl elution buffer (1% (w/v) SDS and 0.1 M Na-bicarbonate), crosslinking reversed by the addition of 0.2 M NaCl, and incubation at 65 °C overnight with agitation (300 rpm). Protein was digested with 20 μg proteinase K for 4 h at 45 °C. RNase A was added for 30 min at 37 °C and DNA was purified using the DNA minielute kit. DNA underwent qPCR analysis, and results were expressed relative to input material.

## siRNA-mediated depletion

HeLa or A549 cells were transfected with siRNA SMARTpools for USP43 (Dharmacon, L-023019-00-0005), USP43 siRNA (Eurofins) (5′-GAAGAUGGUUGCAGAGGAA-3′), MISSION siRNA Universal Negative Control (SIC002, Merck), Scrambled negative control (Eurofins): (5′-CAGUCGCGUUUGCGACUGG-3′), or 14-3-3 PAN siRNA (5′-AAGCTGGCCGAGCAGGCTGAGCGATA-3′) (Dar et al, 2014) using Lipofectamine RNAi MAX. Cells were harvested after 48–72 h for further analysis by flow cytometry, qPCR or immunoblot.

## USP43 transfection or transduction

pHRSIN-pSFFV-pPGK-Puro/Hygro USP43 expressing vector was transfected to $5 \times 10^5$ cells in six-well plates using Fugene. Alternatively, lentivirus was harvested from the transfected cells, as described above. A calcium phosphate ($CaPO_4$) transfection protocol was used to overexpress HA-USP43 constructs for DUB activity assay. HEK293T cells at 40% confluency in 10 ml of DMEM were transfected with 4 μg of plasmid. $CaCl_2$/HBS/DNA precipitate was prepared by mixing 438 μl of $H_2O$ with 2 M $CaCl_2$ and DNA in one tube and then adding it dropwise to 500 μl of 2xHBS (0.27 M NaCl, 1.5 mM $Na_2HPO_4 \cdot 7H_2O$, 0.055 M HEPES, pH 7). The mix was transferred to cells by distributing dropwise all around the dish and gently swirling the dish around. The media was changed the following day and cells were harvested 48 h after transfection.

## DUB activity probe assay

HA-USP43, HA-USP43$^{C110A}$ or HA$^{C110A+H667R}$ were overexpressed in HEK293T cells using the calcium phosphate transfection protocol. Cells were scraped in 500 μl of 1% Triton in 1xPBS, and centrifuged for 10 min at $14,000 \times g$ at 4 °C. Input samples were taken at this point. The remaining samples were precleared with 50 μl sepharose beads for 2 h at 4 °C, and incubated with 25 μl of anti-HA magnetic beads (at 4 °C overnight. The beads were washed three times in 1% Triton in 1xPBS and once in 1xPBS, resuspended in 100 μl of 1xPBS with 5 μl of DUB catalytic activity-based probe Biotin-ANP-Ub-PA (1 μM final concentration) and incubated at 37 °C for 5 min. The reactions were washed three times in 1xPBS, lysed with 50 μl 2 x SDS and analysed by immunoblotting.

## Cycloheximide chase assays

HeLa USP43 KO or overexpressing cells were seeded at $3 \times 10^5$ cells in six-well plates the night before. The cells were pre-incubated in 1% oxygen for 4 h and then treated with 100 μg/ml of cycloheximide for 0, 20. 40 and 60 min and analysed by immunoblotting for HIF-1α degradation.

## Phosphorylation-dependent electrophoretic mobility shift (PDEMS) assay

About $1 \times 10^6$ of Hela cells were washed in 1x TBS buffer (50 mM Tris-Cl, pH 8, 150 mM NaCl), scraped and lysed in 80 µl of RIPA. The lysate was incubated with 1 µl (400 units) of Lambda protein phosphatase (NEB, P0753) according to manufacturer's instructions. The reaction was stopped with 0.2 µl of 0.5 M EDTA, and samples were prepared for immunoblotting by adding 20 µl 6 x SDS loading dye and boiling.

## Immunofluorescence

HeLa cells were seeded on FCS precoated coverslips in 24-well plates at $5 \times 10^4$ density. The following day, cells were incubated in 21 or 1% oxygen for 6 h, washed twice in 1x DPBS and fixed with 4% paraformaldehyde for 15 min. Cells were washed twice in 1x DPBS, permeabilised with 0.1% Triton for 10 min, washed again, and blocked in 4% FBS in 1x DPBS. The coverslips were stained with 1:100 of anti-USP43, or anti-HIF-1α primary antibodies at 4 °C overnight, then washed five times with 1x DPBS and counterstained with 1:400 anti-mouse 647 nm, or anti-rabbit 488 nm Alexa-Fluor™ secondary antibodies for 1.5 h at room temperature. The cells were washed five times in 1x DPBS and mounted in glycerol-based mountant with DAPI. The cells were imaged using LSM 980 with an Airyscan confocal microscope and analysed blinded with Fiji (ImageJ-win64).

## Apoptosis assay

Cells were incubated in 21% oxygen or in 1% oxygen for 6 and 24 h. Cells were harvested by trypsinisation and stained with CellEvent™ Caspase-3/7 Detection Reagent (Invitrogen™, C10423) according to the manufacturer's instructions. Cells were analysed using BD Fortessa (GFP). Cells were treated with 100 µM of Camptothecin (Merck) for 24 h as a control.

## Cell proliferation assay

Cells were seeded in six-well plates at $5 \times 10^4$ density and grow under standard conditions for 24 h. The cells were then incubated in 21 or 1% oxygen for 24, 48 and 72 h.

## Statistical analyses

Quantification and data analysis of experiments are expressed as mean ± sd and *P* values were calculated using analysis of variance (ANOVA) or two-tailed Student's *t*-test for pairwise comparisons using Graphpad Prism v.8. Qualitative experiments were repeated independently to confirm accuracy. Statistical analyses for the CRISPR screens, RNA-seq and ChIP-seq are described in the relevant sections.

# Data availability

The sequencing data from this publication have been deposited to the NCBI Gene Expression Omnibus (https://www.ncbi.nlm.nih.gov/geo/) and assigned the identifiers GSE252651 (CRISPR screens) and GSE252654 (RNA-seq).

The source data of this paper are collected in the following database record: biostudies:S-SCDT-10_1038-S44318-024-00166-6.

# Peer review information

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

## Acknowledgements

We thank all members of the Nathan lab for their helpful comments on the work and manuscript. This work was supported by a Pfizer ITEN award to JAN (Pfizer Inc), a Wellcome Senior Clinical Research Fellowship to JAN (215477/Z/19/Z), and a Lister Institute Research Fellowship to JAN. This work made use of the CIMR FACS facility and was also supported by the NIHR BRC.

## Author contributions

**Tekle Pauzaite**: Conceptualisation; Data curation; Formal analysis; Validation; Investigation; Visualisation; Methodology; Writing—original draft; Writing—review and editing. **Niek Wit**: Resources; Data curation; Software; Formal analysis; Investigation; Methodology; Writing—original draft; Writing—review and editing. **Rachel V Seear**: Formal analysis; Investigation; Methodology. **James A Nathan**: Conceptualisation; Resources; Data curation; Software; Formal analysis; Supervision; Funding acquisition; Validation; Investigation; Visualisation; Methodology; Writing—original draft; Project administration; Writing—review and editing.

In addition to the CRediT author contributions listed above, the contributions in detail are:

Conceptualisation, TP and JAN; Methodology, TP, NW, RVS and JAN; Investigation, TP, NW, RVS and JAN; Writing—original draft, TP and JAN; Writing – reviewing and editing, all authors; Funding acquisition, JAN; Resources, JAN; Supervision, JAN.

Source data underlying figure panels in this paper may have individual authorship assigned. Where available, figure panel/source data authorship is listed in the following database record: biostudies:S-SCDT-10_1038-S44318-024-00166-6.

## Disclosure and competing interests statement

JAN received a Pfizer ITEN discovery grant to fund this work. The remaining authors declare no competing interests.

# Expanded View Figures

**Figure EV1.   Validation of DUBs identified as regulators of the HIF response.**

(**A**) Mixed KO populations of USP43 or USP52 HeLa HRE-$^{ODD}$GFP reporter cells were generated by lentiviral transduction with sgRNA. Cells were then incubated in 1% oxygen for 24 h and analysed by flow cytometry. Representative of three biological replicates. (**B**) HeLa HRE-$^{ODD}$GFP reporter cells transduced with three different sgRNAs against OTUD5 cultured in 21% oxygen. Representative of three biological replicates. (**C–H**) Schematic of the HIF activator validation screen (**C**). The top ten hits by log2 fold change (**H**) were validated in HKC-8 and RPE-1 cells. The cells were transduced with two sgRNAs per DUB. After 8 days, cells were incubated in 1% oxygen for 24 h and analysed by flow cytometry for endogenous cell surface CA9 (**D**, **F**) and intracellular HIF-1α levels (**E**, **G**). DUBs that were validated in either HKC-8, RPE-1 or both cells are highlighted in red (**H**). $n = 4$ biologically independent samples for CA9 levels, and $n = 3$ for HIF-1α levels, mean ± sd. (**I**) Principal component analysis (PCA) plot of RNA-seq of HeLa control, HIF1β KO and USP43 KO cells that were treated with 21 or 1% oxygen ($O_2$) for 16 h before RNA was extracted and sequenced using Hiseq. $n = 3$ biologically independent samples, mean ± sd. (**J**) Quantitative RT-PCR (qPCR) of *USP43* mRNA in A549, MCF7 or HKC-8 cells. Cells were incubated in 1 or 21% oxygen for 16 h. $n = 3$ biologically independent samples. Mean ± sd. **$P = 0.005$, unpaired *t*-test. Source data are available online for this figure.

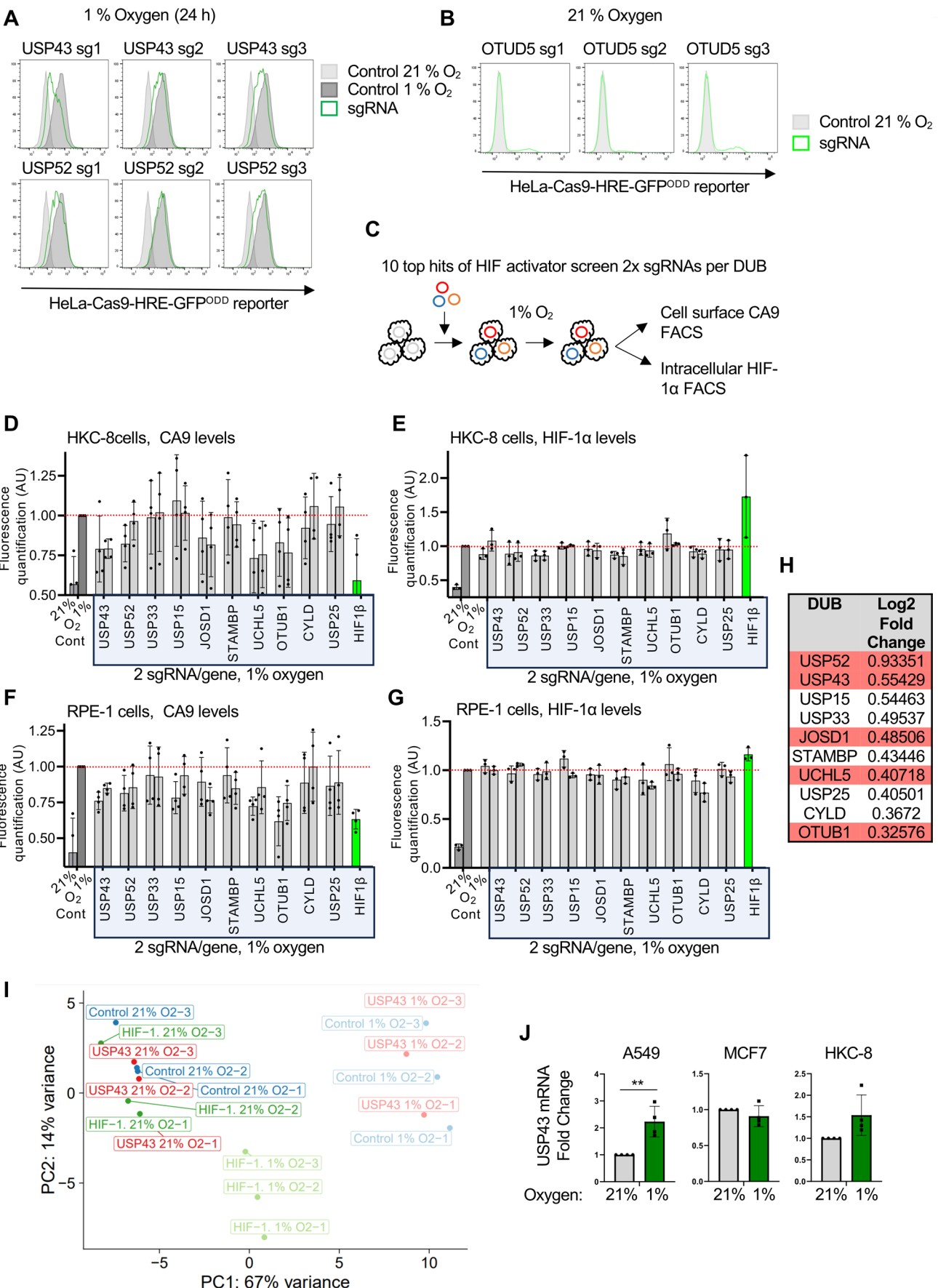

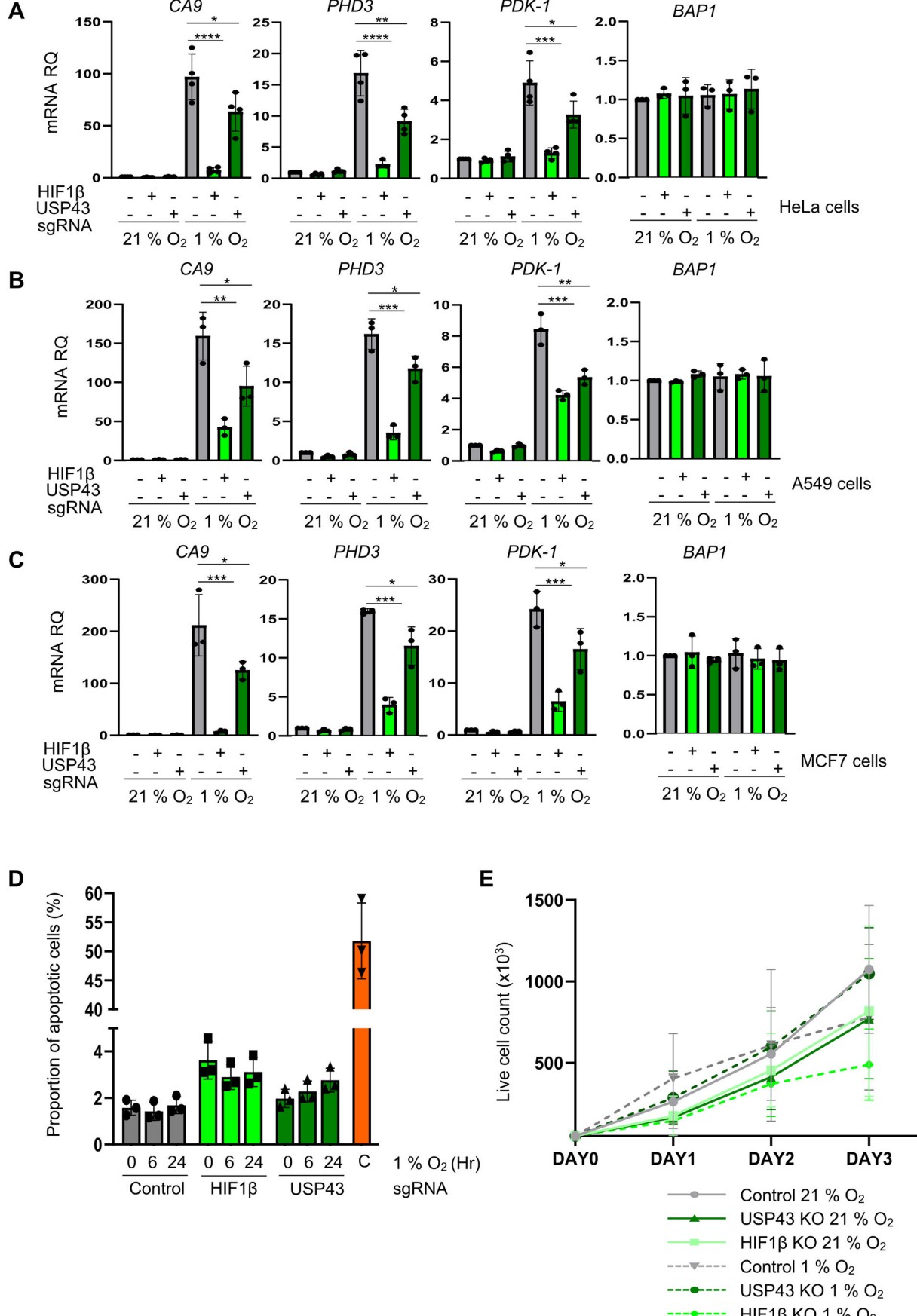

◀  **Figure EV2.  USP43 depletion decreases the expression of selected HIF target genes in hypoxia.**

(A–C) RT-qPCR of *CA9*, *PHD3*, *PDK1*, and *BAP1* in HeLa (A), A549 (B) and MCF7 (C) cells with or without depletion of HIF1β or USP43. Cells were incubated in 21 or 1% oxygen for 16 h prior to lysis. $n = 4$ biologically independent samples, mean ± sd. (A) *CA9*: control vs. HIF1β sgRNA 1% $O_2$ ****$P < 0.0001$, control vs. USP43 sgRNA 1% $O_2$ *$P = 0.0363$. *PHD3*: control vs. HIF1β sgRNA 1% $O_2$ ****$P < 0.0001$, control vs. USP43 sgRNA 1% $O_2$ **$P = 0.0026$. *PDK-1*: control vs. HIF1β sgRNA 1% $O_2$ ***$P = 0.0002$, control vs. USP43 sgRNA 1% $O_2$ *$P = 0.0299$. One-way ANOVA. (B) $n = 3$ biologically independent samples, mean ± sd. *CA9*: control vs. HIF1β sgRNA 1% $O_2$ **$P = 0.0018$, control vs. USP43 sgRNA 1% $O_2$ *$P = 0.0295$. *PHD3*: control vs. HIF1β sgRNA 1% $O_2$ ***$P = 0.0001$, control vs. USP43 sgRNA 1% $O_2$ *$P = 0.0233$; PDK-1: control vs. HIF1β sgRNA 1% $O_2$ ***$P = 0.0004$, control vs. USP43 1% $O_2$ **$P = 0.0023$. One-way ANOVA. (C) $n = 3$ biologically independent samples, mean ± sd. *CA9*: control vs. HIF1β sgRNA 1% $O_2$ ***$P = 0.0008$, control vs. USP43 sgRNA 1% $O_2$ *$P = 0.0433$. *PHD3*: control vs. HIF1β sgRNA 1% $O_2$ ***$P = 0.0001$, control vs. USP43 sgRNA 1% $O_2$ *$P = 0.0209$. *PDK-1*: control vs. HIF1β sgRNA 1% $O_2$ ***$P = 0.0009$, control vs. USP43 sgRNA 1% $O_2$ *$P = 0.0454$. One-way ANOVA. (D) Apoptosis assay. HeLa control (grey), HIF1β depleted (light green) or USP43 depleted (dark green) cells were incubated in 1% oxygen for 0, 6 or 24 h, and apoptosis was measured using CellEvent™ Caspase-3/7 Green Flow Cytometry Assay Kit (C10427). Camptothecin (C) (100 μM, 24 h) was used as a positive control. $n = 3$ biological replicates, mean ± sd. (E) Cell growth assay. HeLa control (grey), HIF1 depleted (light green) or USP43 depleted (dark green) cells were incubated in 21 (solid line) or 1% (dotted line) oxygen for 0, 24, 48 and 72 h and counted at each time point. The cell growth plot represents three biologically independent experiments, mean ± sd. Source data are available online for this figure.

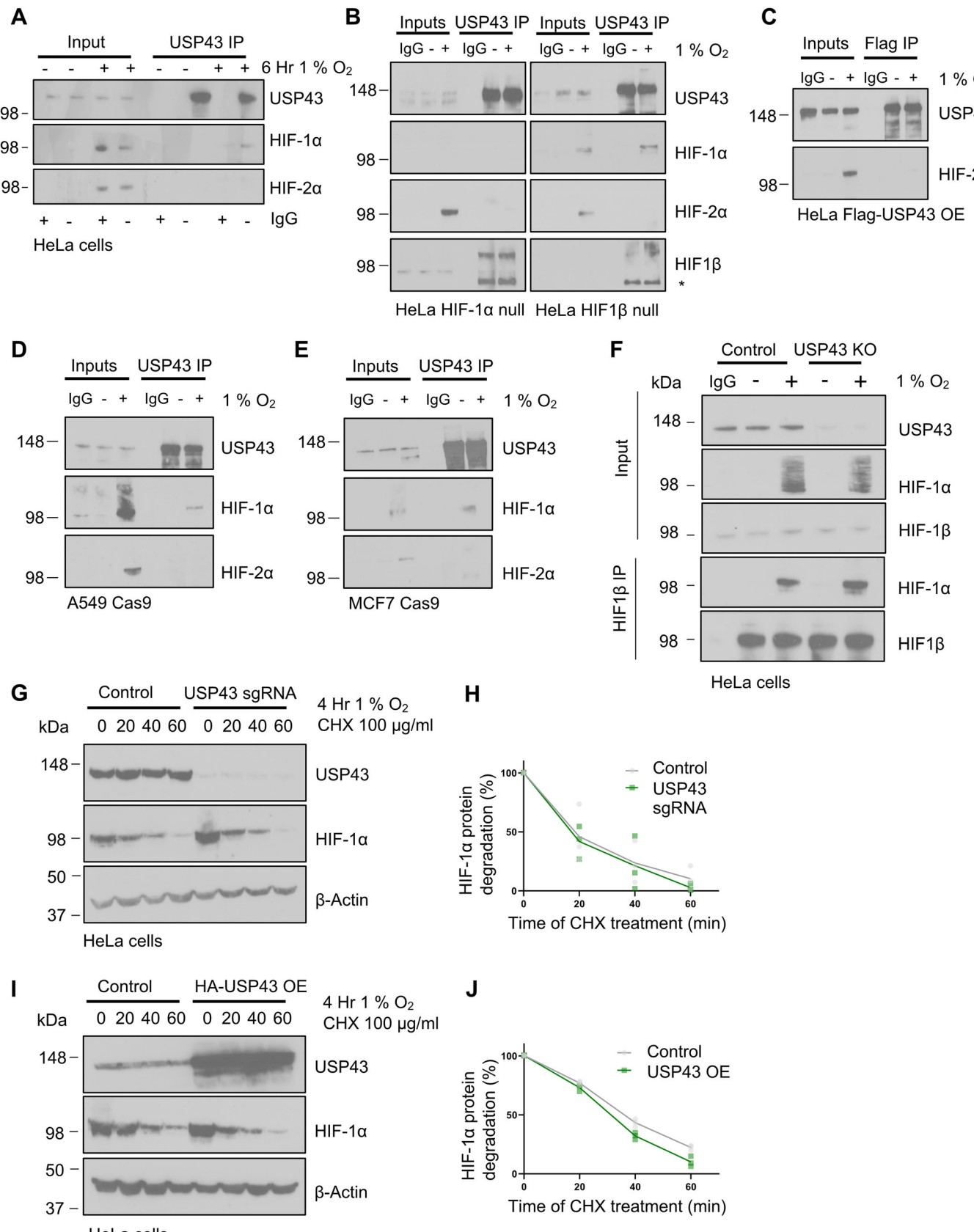

◀   **Figure EV3.   USP43 interacts with HIF-1α but not HIF-2α.**

(A) Endogenous USP43 was immunoprecipitated in HeLa cells grown in 21 or 1% oxygen for 6 h. Samples were immunoblotted for HIF-1α and HIF-2α. Representative of three biological replicates. (B) Endogenous USP43 was immunoprecipitated in HIF-1α or HIF1β clonal KO HeLa cells grown in 21 or 1% oxygen for 6 h. Samples were immunoblotted for HIF-1α, HIF-2α and HIF1β. Representative of three biological replicates. (C) USP43-Flag was immunoprecipitated in USP43-Flag overexpressing HeLa cells grown in 21 or 1% oxygen for 6 h. Samples were immunoblotted for HIF-2α. Representative of three biological replicates. (D, E) Endogenous USP43 was immunoprecipitated in A549 (D) or MCF7 (E) cells grown in 21 or 1% oxygen for 6 h. Samples were immunoblotted for HIF-1α and HIF-2α. Representative of three biological replicates. (F) Endogenous HIF1β was immunoprecipitated in HeLa and USP43 depleted cells grown in 21 or 1% oxygen for 6 h. Samples were immunoblotted for HIF-1α. Representative of three biological replicates. (G, H) Control or mixed population USP43 KO HeLa cells were incubated in 1% oxygen for 4 h and then treated with cycloheximide (100 μg/ml) in hypoxia for 0–60 min. HIF-1α levels were measured by immunoblot (G) and quantified by densitometry using ImageJ ($n = 3$) (H). (I, J) Control or USP43 overexpressing (OE) HeLa cells were incubated in 1% oxygen for 4 h and then treated with cycloheximide (100 μg/ml) in hypoxia for 0–60 min. HIF-1α levels were measured by immunoblot (I) and quantified by densitometry using ImageJ ($n = 3$) (J). Source data are available online for this figure.

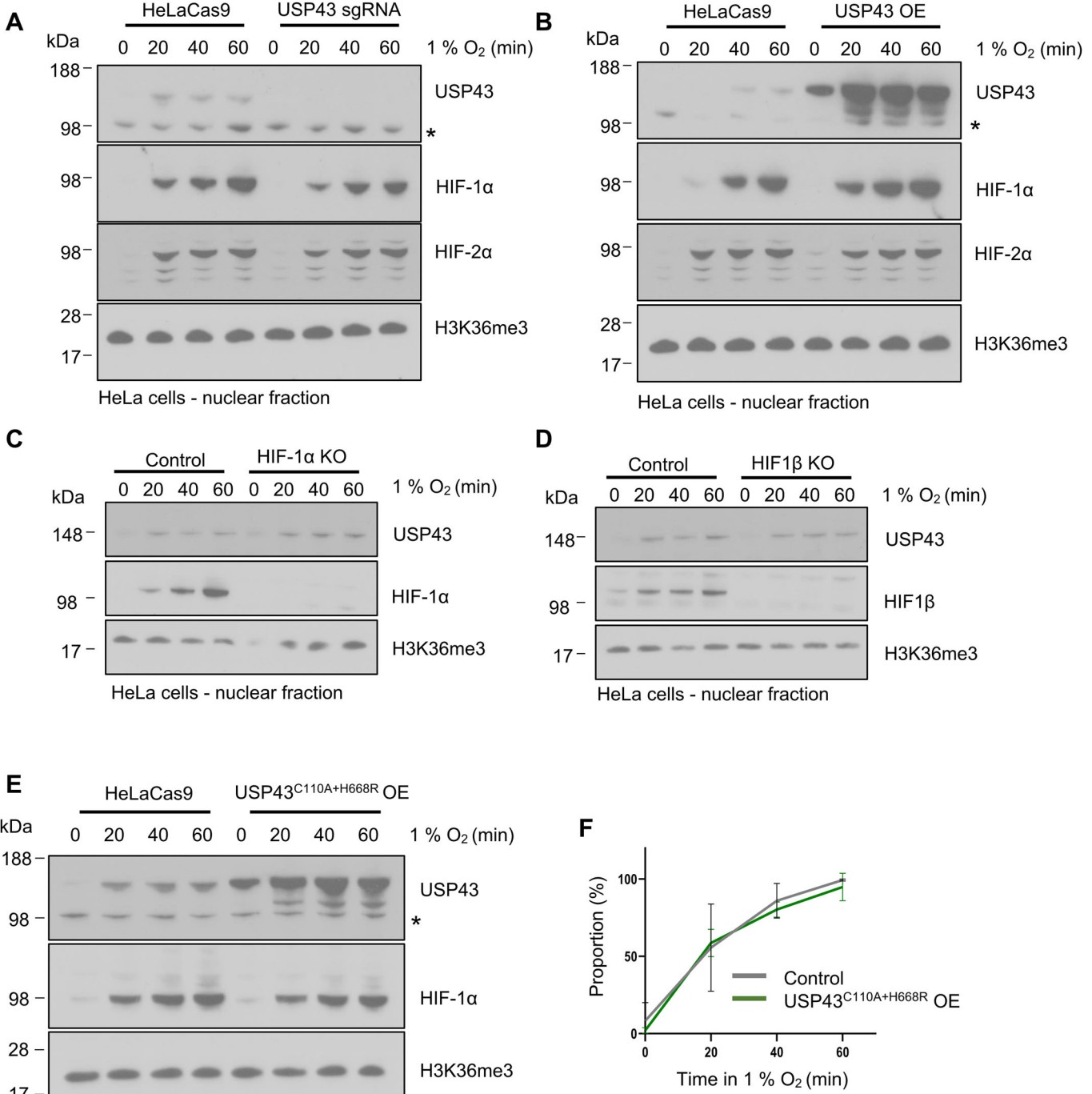

**Figure EV4.  USP43 regulates HIF-1α nuclear accumulation.**

(**A, B**) Immunoblot of the nuclear fraction of HeLa-Cas9 or mixed population USP43 KO cells (**A**), USP43 overexpressing (OE) cells (**B**). (**A**) shows an extended immunoblot of Fig. 5C. Representative of three biological replicates. *non-specific band. (**C**) Immunoblot of the nuclear fraction of HeLa-Cas9 or HIF-1α KO cells incubated in 1% oxygen for 0–60 min. Samples were immunoblotted for USP43, HIF-1α and H3K36me3 as a loading control. Representative of three biological replicates. (**D**) As for (**C**) but in HIF1β KO cells. Representative of three biological replicates. (**E, F**) Immunoblot of the nuclear fraction of HeLa-Cas9 or USP43$^{C110A+H668R}$ OE cells (**E**) incubated in 1% oxygen for 0–60 min. Representative of three biological replicates. *non-specific band. Quantification of HIF-1α enrichment within the nuclear fraction in HeLa control or USP43$^{C110A+H668R}$ OE cells (**F**). HIF-1α levels relative to a stable histone mark (H3K36me3) were measured by immunoblot. $n = 3$ biologically independent samples, mean ± sd. Source data are available online for this figure.

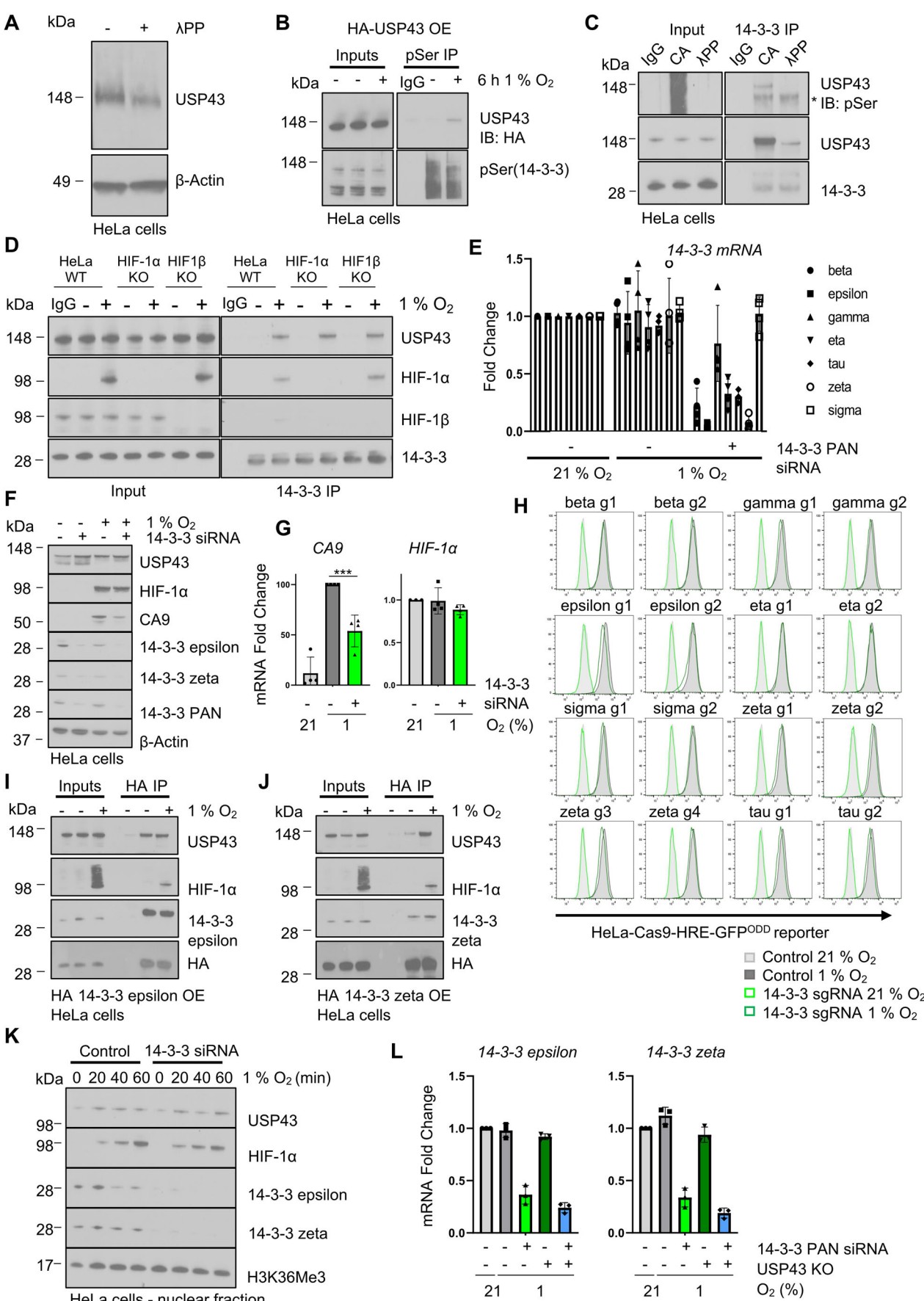

**Figure EV5.  USP43 associates with 14-3-3 proteins to regulate HIF-1 signalling.**

(A) Immunoblot of the phosphorylation-dependent electrophoretic mobility shift (PDEMS) assay of USP43, with or without lambda protein phosphatase (λPP) treatment. (B) Endogenous pSer(14-3-3 motif) was immunoprecipitated in HA-USP43 overexpressing HeLa cells incubated in 21 or 1% oxygen for 6 h. Representative of three biological replicates. (C) Endogenous 14-3-3 was immunoprecipitated in HeLa cells with or without Calyculin A (CA) treatment (100 nM, 37 °C for 30 min) or λPP (400 units, 30 °C for 30 min). Samples were immunoblotted for pSer(14-3-3 motif), USP43, and 14-3-3. Representative of three biological replicates. (D) Endogenous 14-3-3 was immunoprecipitated in wildtype, HIF-1α or HIF1β clonal KO HeLa cells grown in 21 or 1% oxygen for 6 h. Samples were immunoblotted for USP43, HIF-1α and HIF1β. Representative of three biological replicates. (E) qPCR of HeLa cells transfected with a 14-3-3 pan siRNA and incubated in 1 or 21% oxygen for 16 h. Expression of the seven *14-3-3* isoforms were analysed using individual primer pairs. $n = 3$ biologically independent samples, mean ± sd. (F) Immunoblot of control or 14-3-3 siRNA-depleted HeLa cells incubated in 21 or 1% oxygen for 16 h. Representative of three biological replicates. (G) qPCR of *CA9* and *HIF-1α* mRNA expression control or 14-3-3 siRNA-depleted HeLa cells incubated in 21 or 1% oxygen for 16 h. $n = 3$ biologically independent samples, mean ± sd. ***$P = 0.0002$, one-way ANOVA. (H) Mixed KO populations of HeLa HRE-[ODD]GFP reporter cells to each 14-3-3 isoform were generated by lentiviral transduction of sgRNA (two sgRNAs for each isoforms, four sgRNAs for zeta isoform). Cells were incubated in 21 or 1% oxygen for 16 h and analysed by flow cytometry. (I, J) HeLa cells were transfected with HA-14-3-3 epsilon (I) or HA-14-3-3 zeta (J) and incubated in 21 or 1% oxygen for 6 h. The 14-3-3 isoforms were immunoprecipitated using the HA tag and immunoblotted for USP43. Representative of three biological replicates. (K) HIF-1α enrichment within the nuclear fraction in HeLa control or following 14-3-3 siRNA-mediated depletion. Cells were incubated in 1% oxygen for 0 to 60 min. (L) qPCR of *14-3-3 epsilon* and *zeta* expression in control or USP43 null cells, with or without 14-3-3 siRNA-mediated depletion. Cells were incubated in 21 or 1% oxygen for 6 h prior to lysis. $n = 3$, biologically independent samples, mean ± sd. Source data are available online for this figure.

