## [Peer Review File · The EMBO Journal]

Deubiquitinating enzyme mutagenesis screens identify a USP43-dependent HIF-1 transcriptional response

Tekle Pauzaite, Niek Wit, Rachel Seear, and James Nathan

Corresponding author(s): James Nathan (jan33@cam.ac.uk)

Review Timeline:

Submission Date:	9th Jan 24
Editorial Decision:	26th Feb 24
Revision Received:	13th May 24
Accepted:	24th Jun 24

Editors: Hartmut Vodermaier and William Teale

Transaction Report:

Prof. James A Nathan
Cambridge Institute of Therapeutic Immunology & Infectious Disease, University of Cambridge
Department of Medicine
Hills Road
Cambridge, Cambridgeshire CB2 0XY
United Kingdom

26th Feb 2024

Re: EMBOJ-2024-116611
Deubiquitinating enzyme mutagenesis screens identify a USP43 driven HIF-1 transcriptional response

Dear James,

Thank you again for submitting your study on USP43/HIF interplay, and once more apologies for the delay in getting back to you with the results of its evaluation. We have now finally received the reports from all three referees that had agreed to review the study. As you will see from the comments copied below, all reviewers consider your findings of interest and the study generally well-conducted. Following adequate addressing of several specific queries raised by each of the referees, we would therefore be happy to consider a revised manuscript further for publication in The EMBO Journal.

Since our policy to allow only a single round of major revision makes it important to satisfactorily answer to all comments at the time of resubmission, I would be happy to look at a tentative response letter/revision plan already during the early stages of the revision work, in order to discuss how to best address the more major issues. I should add that we could also offer extension of the default three-months revision period if needed, with our 'scooping protection' (meaning that competing work appearing elsewhere in the meantime will not affect our considerations of your study) remaining of course valid also throughout this extension.

Detailed information on preparing, formatting and uploading a revised manuscript can be found below and in our Guide to Authors. Thank you again for the opportunity to consider this work for The EMBO Journal, and I look forward to hearing from you in due time.

With kind regards,

Hartmut

9) Digital image enhancement is acceptable practice, as long as it accurately represents the original data and conforms to community standards. If a figure has been subjected to significant electronic manipulation, this must be clearly noted in the figure legend and/or the 'Materials and Methods' section. The editors reserve the right to request original versions of figures and the original images that were used to assemble the figure. Finally, we generally encourage uploading of numerical as well as gel/blot image source data; for details see: embopress.org/page/journal/14602075/authorguide#sourcedata

At EMBO Press, we ask authors to provide source data for the main manuscript figures. Our source data coordinator will contact you to discuss which figure panels we would need source data for and will also provide you with helpful tips on how to upload and organize the files.

In the interest of ensuring the conceptual advance provided by the work, we recommend submitting a revision within 3 months (26th May 2024). Please discuss the revision progress ahead of this time with the editor if you require more time to complete the revisions. Use the link below to submit your revision:

Link Not Available

Referee #1:

In this manuscript, Pauzaite et al. have carried out enzyme mutagenesis screens to investigate the role of DUBs in HIF signalling. They identify USP43 in the control of HIF-dependent gene expression and demonstrate that it regulates the nuclear accumulation, rather than the stability of HIF-1alpha. Mechanistically, this hypoxia-dependent association with 14-3-3 proteins. The authors then claim that their data "unveils the DUB landscape in HIF signalling". While a significant amount of work has been done in this area, it remains an area of interest and some interesting new mechanistic data on USP43 in the regulation of HIF is presented. The experimental work is interesting and of high quality but the novelty of the central question is somewhat overstated as a significant amount of work on DUBs and HIF has been published to date.

Major Points:

1) While the role of DUBs is indeed a continuing area of interest, the novelty of the current study is somewhat overstated as multiple papers have identified roles for specific DUBs in the regulation of the HIF pathway. Some papers have not been cited (e.g. PMID: 35191554; PMID: 35440539; PMID: 36656861; PMID: 25615526).

2) While the authors do investigate the effects of USP43 on endogenous HIF and in other cell lines, two weaknesses of the screening assay in Fig 1 are 1) its reliance on an overexpressed reporter and 2) the fact that it is done in HeLa cells which are a poor representation of anything approaching a "normal" cell. Verification of the screening assay depicted in figure one in a more physiologically relevant model would help.

3) In Figure 2, Hypoxia experiments should include normoxic controls and Roxadustat experiments should have vehicle controls.

4) Does endogenous USP43 directly interact with endogenous HIF-1alpha and not HIF-2alpha in a hypoxia / PHI dependent manner in multiple cell types? This is important to demonstrate.

Referee #2:

In this study, Pauzaite et al investigate novel regulators of Hypoxia Inducible Factor using a CRISPR screen. They identify and focus on USP43, as a DUB required for full HIF activity. They show defects in nuclear accumulation and chromatin binding but no changes to mRNA or protein levels of HIF, and the effects are only on HIF-1, not HIF-2. Overall, this is a novel and interesting study, covering an aspect that is still underinvestigated, DUBs and hypoxia.

The functional data is very convincing and it is clear USP43 is altering HIF-1 function. I have a few specific questions that I think would help define the mechanism proposed.

For example, does overexpression of USP43 lead to increased HIF nuclear accumulation? Is this activity dependent? Although rescue experiments are present, showing the system is plastic (Implied by increased mRNA) would be required.

Does lack of USP43 prevent HIF-1a HIF-1b dimer formation?

Does blocking nuclear import resolve the defect created by USP43 or 14-3-3 depletion?

Does 14-3-3 bind to HIF-1a or HIF-1 or the dimer?

Minor point:

How common is USP43 mRNA upregulation in hypoxia. Analysing the 3 cell lines used in the study would be sufficient.

There is a big gap between 14-3-3/usp43 involvement and nuclear translocation, this should be discussed a bit more in the discussion.

Referee #3:

In this study, Nathan and colleagues performed a pooled DUB CRISPR screen in HeLa cells aiming to identify DUBs that regulate the HIF response to hypoxia. They observed that loss of USP43 reduced expression of a subset of HIF-1 target genes without affecting HIF-1 stability. Specifically, USP43 depletion impaired the HIF-driven shift to glycolysis. Interestingly, catalytic activity of USP43 was not essential for the effect of USP43 on HIF-mediated transcription. The authors demonstrate that USP43 stimulates localization of HIF-1 at chromatin via the interaction of 14-3-3 proteins.

Taken together, the presented work reveals an additional control layer of the hypoxia response pathway, it delineates the molecular details how USP43 promotes specific HIF-1 target gene expression independently of its catalytic DUB activity.

In general, the presented experiments are well designed, carefully controlled and adequately replicated. Complementary approaches were used to substantiate the individual findings. The statistical analyses are adequate and the conclusions drawn based on the presented data are justified. The data and the methods are presented clearly and with sufficient detail. Publication should be considered after addressing the following comments:

- Fig. 3 A/B: Upon HIF1-beta or USP43 knockdown, was the HIF reporter activity measures in 1% or 21% oxygen?

- To better judge the overall impact of USP43-controlled HIF-1 positioning at specific target gene promoters, cell viability/apoptosis induction should be determined upon hypoxia/ 1% oxygen treatment (6 h, 24 h). How do cells cope with prolonged hypoxia (>24 h)? Does USP43 depletion has a significant effect on how cells survive under hypoxic conditions?

Minor comment:

- Fig. 2: On page 5, the authors write "To substantiate the involvement of USP43 in HIF-transcriptional activation of target genes, we undertook RNA-seq analysis in HeLa control, USP43 depleted, or HIF1b depleted cells following 6 hr of incubation in 21 % or 1 % oxygen (Fig 2G; S1C)." The RNA-seq analysis is shown in figure 2F.

Response to Reviewers' Comments

Referee #1:

In this manuscript, Pauzaite et al. have carried out enzyme mutagenesis screens to investigate the role of DUBs in HIF signalling. They identify USP43 in the control of HIF-dependent gene expression and demonstrate that it regulates the nuclear accumulation, rather than the stability of HIF-1 α . Mechanistically, this hypoxia-dependent association with 14-3-3 proteins. The authors then claim that their data "unveils the DUB landscape in HIF signalling". While a significant amount of work has been done in this area, it remains an area of interest and some interesting new mechanistic data on USP43 in the regulation of HIF is presented. The experimental work is interesting and of high quality but the novelty of the central question is somewhat overstated as a significant amount of work on DUBs and HIF has been published to date.

We thank the Reviewer for their appreciation of our studies. We agree that other DUBs have been implicated in the HIF pathway. Our unbiased approach to explore VHL antagonism and other aspects of HIF pathway builds on this prior body of work. To reflect this Reviewer's point, we have rephrased our manuscript to acknowledge prior studies more clearly (see Introduction and Discussion). We also highlight where there are differences in interpretation (e.g. the requirement for VHL antagonism), HIF- α isoform specificity, and the role of non-catalytic functions.

Major Points:

1) While the role of DUBs is indeed a continuing area of interest, the novelty of the current study is somewhat overstated as multiple papers have identified roles for specific DUBs in the regulation of the HIF pathway. Some papers have not been cited (e.g. PMID: 35191554; PMID: 35440539; PMID: 36656861; PMID: 25615526).

Thank you for highlighting these papers, which have now been included. As noted, we do acknowledge the contributions of prior studies but many questions remain regarding the evidence for VHL antagonism when oxygen is not limiting, the use of over-expression systems for DUB activity assays, and the need to consider non-catalytic functions of DUBs. This is the first study, to our knowledge, to globally assess DUB function in hypoxia and HIF assays. Prior studies have used selected siRNA, or arrayed shRNA libraries that only contain a relatively focused panel of DUBs. As a result, our findings make several important contributions, including the identification of USP43, the involvement of 14-3-3, the importance of non-catalytic functions of DUBs, and that we do not find strong evidence for DUBs antagonising VHL when oxygen is not limiting.

2) While the authors do investigate the effects of USP43 on endogenous HIF and in other cell lines, two weaknesses of the screening assay in Fig 1 are 1) its reliance on an overexpressed reporter and 2) the fact that it is done in HeLa cells which are a poor representation of anything approaching a "normal" cell. Verification of the screening assay depicted in figure one in a more physiologically relevant model would help.

We view the reporter as a strength of the assay. It can only be activated by endogenous HIF and the fusion of GFP to the ODD domain of HIF-1 α provides a sensitive readout for VHL-mediated ubiquitination. The reporter also allows iterative flow cytometry sorting in live cells, which cannot be achieved by other methods. In line with this, the reporters have been used by several groups to both monitor HIF activity and to understand how it is regulated. HeLa cells have also provided a robust cell model to uncover key aspects of HIF signalling (e.g. PMID 11595184, 10878807), the ubiquitin-proteasome pathway (e.g. 12648459, 17443180), and many other essential biological processes that occur in both cancerous and primary cells.

To experimentally address the points raised about the choice of cell line and reporter, we have undertaken further verification of the screen, as suggested. We undertook sgRNA-mediated depletion of the top 10 hits (based on fold change) in two non-cancerous models, retinal pigment epithelial cells (RPE1s) and human kidney proximal renal tubular cells (HKC-8). We measured the effect of depletion of these DUBs on endogenous HIF-1 α and its endogenous target gene, CA9 (**New Figure EV1C-H**), avoiding any reliance on a reporter system. USP43 and USP52, the top hits in the screen reduced CA9 levels in both cell types. Three of the other top hits (JOSD1, UCHL5 and OTUB1), which were below our statistical threshold in the HeLa screen, also validated. Thus, the validation assays confirm the findings from our reporter screen. We also highlight that our identification of USP43-mediated regulation of HIF signalling is supported by our observations in 293T, RCC4, 786-O, MCF7 and A549 cells.

3) In Figure 2, Hypoxia experiments should include normoxic controls and Roxadustat experiments should have vehicle controls.

The 21 % oxygen controls have now been included (**New Figure 2A**). The Roxadustat experiments did have DMSO as a vehicle control, but to make this clearer we have included a further experiment in A549 cells (**New Figure 2H**).

4) Does endogenous USP43 directly interact with endogenous HIF-1 α and not HIF-2 α in a hypoxia / PHI dependent manner in multiple cell types? This is important to demonstrate.

We now include endogenous USP43 immunoprecipitation experiments in A549 and MCF7 cells in hypoxia (**New Figure EV3D, E**). We also show that USP43 does not bind HIF-2 α when HIF-1 α or HIF1 β are depleted (**New Figure EV3B**). Even when USP43 is overexpressed we still do not see an association with HIF-2 α (**New Figure EV3C**). These findings are consistent with the HIF-1 α specific effect observed in RCC4 cells but not 786-O cells. They also confirm that the association is specific to HIF-1 α and not just dependent on the relative abundance of the HIF- α forms.

Referee #2:

In this study, Pauzaitė et al investigate novel regulators of Hypoxia Inducible Factor using a CRISPR screen. They identify and focus on USP43, as a DUB required for full HIF activity. They show defects in nuclear accumulation and chromatin binding but no changes to mRNA or protein levels of HIF, and the effects are only on HIF-1, not HIF-2. Overall, this is a novel and interesting study, covering an aspect that is still underinvestigated, DUBs and hypoxia. The functional data is very convincing and it is clear USP43 is altering HIF-1 function. I have a few specific questions that I think would help define the mechanism proposed.

For example, does overexpression of USP43 lead to increased HIF nuclear accumulation? Is this activity dependent? Although rescue experiments are present, showing the system is plastic (Implied by increased mRNA) would be required.

We thank the Reviewer for their support of our work and helpful suggestions.

We did consider that USP43 overexpression might potentiate HIF-1 α nuclear accumulation and included this in the original manuscript. We observed that USP43 overexpression led to HIF-1 α accumulation by both chromatin fractionation and high salt fractionation **Figure 5G and H**. The catalytic inactive USP43 mutant did not cause HIF-1 α accumulation **Figure EV6E and F**. Therefore, as previous, we speculated that there may be some contribution of USP43 catalytic activity. We have made these points clearer in the revised manuscript.

We agree about the plasticity of the rescue experiments. We now include the corresponding USP43 qPCR data for the reconstitution studies. These nicely demonstrate the plasticity of the overexpression effect (**New Appendix Figure S3B**).

Does lack of USP43 prevent HIF-1a HIF-1b dimer formation?

Thank you for raising this important question. We now include immunoprecipitation experiments of the HIF subunits with or without USP43. We do not observe any effect on dimer formation (**New Figure EV3F**).

Does blocking nuclear import resolve the defect created by USP43 or 14-3-3 depletion?

This is an interesting question that we wanted to investigate. However, as global inhibition of nuclear import would also prevent HIF translocation, these types of inhibitory experiments are not possible. As an alternative strategy, we examined if we could prevent selective USP43 nuclear import.

Although the nuclear import signal for USP43 is not known, there is one putative nuclear localisation signal within USP43 (aa 15-26, APRPRRRRSLRRL). We mutated this region (USP43 Δ^{2-26}) and compared the localisation to wildtype USP43 (**Rebuttal Figure 1**). USP43 was still observed in the chromatin fraction, irrespective of this mutation. It is possible that an alternative region of USP43 may be important, or that it associates with additional factors to enter the nucleus. 14-3-3 proteins may assist USP43 translocation, as 14-3-3 freely shuttles in and out of the nucleus (PMID 11864996), or 14-3-3 ligand binding may alter the relative import/export of USP43 and HIF. While of interest, the further mechanisms of nuclear localisation of HIF-1 α and USP43 are beyond the scope of this manuscript but will be interesting to explore in our ongoing work. We include a more detailed discussion of these points and possible explanations of our findings.

Rebuttal Figure 1: Wildtype or USP43 Δ^{2-26} were overexpressed in USP43 null HeLa cells by lentiviral transduction. Immunoblots of the soluble (S) and chromatin fractions (Ch) obtained using the formaldehyde cross-linking chromatin extraction protocol. H3K36me3 served as a control for the chromatin fraction.

Does 14-3-3 bind to HIF-1a or HIF-1 or the dimer?

We now include experiments in HIF1 β null cells to address this (**New Figure EV5D**). 14-3-3 binds HIF-1 α irrespective of whether it associates with HIF1 β .

Minor point:

How common is USP43 mRNA upregulation in hypoxia. Analysing the 3 cell lines used in the study would be sufficient.

Thank you for this suggestion. We now include qPCR data of USP43 levels in A549, MCF7, and HKC8 cell lines. Interesting, USP43 upregulation in hypoxia occurs in A549 and HKC8 cells but not in MCF7 cells (**New Figure EV1J**). It will be of interest in future studies to explore why MCF7 cells do not respond but HIF target genes are known to vary between cell types.

There is a big gap between 14-3-3/usp43 involvement and nuclear translocation, this should be discussed a bit more in the discussion.

We now include a more detailed discussion exploring the role of 14-3-3 in nuclear import or export, the potential involvement of ligand binding, and how phosphorylation may alter the translocation. This also relates to the earlier point regarding nuclear import.

Referee #3:

In this study, Nathan and colleagues performed a pooled DUB CRISPR screen in HeLa cells aiming to identify DUBs that regulate the HIF response to hypoxia. They observed that loss of USP43 reduced expression of a subset of HIF-1 target genes without affecting HIF-1 stability. Specifically, USP43 depletion impaired the HIF-driven shift to glycolysis. Interestingly, catalytic activity of USP43 was not essential for the effect of USP43 on HIF-mediated transcription. The authors demonstrate that USP43 stimulates localization of HIF-1 at chromatin via the interaction of 14-3-3 proteins.

Taken together, the presented work reveals an additional control layer of the hypoxia response pathway, it delineates the molecular details how USP43 promotes specific HIF-1 target gene expression independently of its catalytic DUB activity.

In general, the presented experiments are well designed, carefully controlled and adequately replicated. Complementary approaches were used to substantiate the individual findings. The statistical analyses are adequate and the conclusions drawn based on the presented data are justified. The data and the methods are presented clearly and with sufficient detail. Publication should be considered after addressing the following comments:

We thank the Reviewer for their appreciation of our work.

- Fig. 3 A/B: Upon HIF1-beta or USP43 knockdown, was the HIF reporter activity measured in 1% or 21% oxygen?

We apologise for this omission which has now been corrected. It was stated in the figure legend in the original manuscript but should have been clearer in the figure itself. Reporter activity was measured in 1% oxygen aside from showing the 21% control. We have clarified this and included the 1% and 21% oxygen flow cytometry measurements (**New Figure 3A, B**). USP43 KO and HIF1 β KO do not alter the reporter in 21%, as shown in Figure 1D.

- To better judge the overall impact of USP43-controlled HIF-1 positioning at specific target gene promoters, cell viability/apoptosis induction should be determined upon hypoxia/ 1% oxygen treatment (6 h, 24 h). How do cells cope with prolonged hypoxia (>24 h)? Does USP43 depletion have a significant effect on how cells survive under hypoxic conditions?

Thank you for these suggestions. We now include experiments examining cell viability and apoptosis induction (**New Figure EV2D, E**). There was no significant effect of USP43 loss on apoptosis at 6 h or 24 h of hypoxia. Similar findings were observed with HIF1 β depletion.

There was also no growth defect in the USP43 cells after prolonged hypoxia exposure (**New Figure EV2E**). This does differ from HIF1 β loss where there is small growth defect (PMID 34125486), and likely relates to USP43 altering the transcriptional activation of a subset of HIF-1 target genes.

Minor comment:

- Fig. 2: On page 5, the authors write "To substantiate the involvement of USP43 in HIF-transcriptional activation of target genes, we undertook RNA-seq analysis in HeLa control, USP43 depleted, or HIF1b depleted cells following 6 hr of incubation in 21 % or 1 % oxygen (Fig 2G; S1C)." The RNA-seq analysis is shown in figure 2F.

This has been corrected.

Dear James,

I am pleased to inform you that your manuscript has been accepted for publication in the EMBO Journal.

Many congratulations to you and your team!

Best wishes,

William

William Teale, PhD
Editor
The EMBO Journal
w.teale@embojournal.org

Referee #2:

The authors have addressed the majority of my concerns very well.

Referee #3:

Nathan and colleagues addressed all comments sufficiently and in great detail. New data was included that further support the drawn conclusions. I can recommend the publication of the manuscript in its current form.
